# Targeted temperature management at 33˚C or 36˚C induces equivalent myocardial protection by inhibiting HMGB1 release in myocardial ischemia/reperfusion injury

Jin Ho Beom[1], Ju Hee Kim[1], Jeho Seo[1], Jung Ho Lee[2,3], Yong Eun Chung[4], Hyun Soo Chung[1], Sung Phil Chung[1], Chul Hoon Kim[2]*, Je Sung You[1]*

1 Department of Emergency Medicine, Yonsei University College of Medicine, Seoul, Republic of Korea, 2 Department of Pharmacology, BK21 PLUS Project for Medical Science, Brain Research Institute, Yonsei University College of Medicine, Seoul, Republic of Korea, 3 Department of Pharmacology, Eulji University School of Medicine, Daejeon, Republic of Korea, 4 Department of Radiology, Yonsei University College of Medicine, Seoul, Republic of Korea

* youjsmd@yuhs.ac (JSY); kimhoon@yuhs.ac (CHK)

**Data Availability Statement:** All relevant data are within the manuscript and its Supporting information files.

## Abstract

Acute myocardial infarction (AMI) is lethal and causes myocardial necrosis via time-dependent ischemia due to prolonged occlusion of the infarct-related artery. No effective therapy or potential therapeutic targets can prevent myocardial ischemia/reperfusion (I/R) injury. Targeted temperature management (TTM) may reduce peri-infarct regions by inhibiting the extracellular release of high mobility group box-1 (HMGB1) as a primary mediator of the innate immune response. We used a rat left anterior descending (LAD) coronary artery ligation model to determine if TTM at 33˚C and 36˚C had similar myocardial protective effects. Rats were divided into sham, LAD I/R+37˚C normothermia, LAD I/R+33˚C TTM, and LAD I/R+36˚C TTM groups (n = 5 per group). To verify the cardioprotective effect of TTM by specifically inhibiting HMGB1, rats were assigned to sham, LAD I/R, and LAD I/R after pre-treatment with glycyrrhizin (known as a pharmacological inhibitor of HMGB1) groups (n = 5 per group). Different target temperatures of 33˚C and 36˚C caused equivalent reductions in infarct volume after myocardial I/R, inhibited the extracellular release of HMGB1 from infarct tissue, and suppressed the expression of inflammatory cytokines from peri-infarct regions. TTM at 33˚C and 36˚C significantly attenuated the elevation of cardiac troponin, a sensitive and specific marker of heart muscle damage, after injury. Similarly, glycyrrhizin alleviated myocardial damage by suppressing the extracellular release of HMGB1. TTM at 33˚C and 36˚C had equivalent myocardial protective effects by similar inhibiting HMGB1 release against myocardial I/R injury. This is the first study to suggest that a target core temperature of 36˚C is applicable for cardioprotection.

**Funding:** This work was supported by the Basic Science Research Program of the National Research Foundation of Korea (NRF), funded by the Ministry of Science and ICT (grant numbers NRF-2015R1C1A1A01054641 and NRF-2018R1C1B6006159 to J.S.Y; NRF-2019R1A2C3002354 to C.H.K; and NRF-2019R1C1C1006332 to J.B.) and a faculty research grant from the Yonsei University College of Medicine (grant number 6-2017-0092 to J.B. and 6-2019-0188 to J.S.Y.). The funding bodies had no role in the design, collection, analysis, or interpretation of this study.

**Competing interests:** The authors have declared that no competing interests exist.

## Introduction

Coronary heart disease is the leading cause of death worldwide, and acute myocardial infarction (AMI) is the most severe manifestation of this disease [1, 2]. In AMI, prolonged occlusion of the infarct-related artery leads to high levels of myocardial necrosis as a time-dependent ischemic process. To minimize myocardial necrosis, blood flow to the infarct-related artery must be restored by mechanical reperfusion using a coronary artery stent and thrombolytic therapy as rapidly as possible [3, 4]. Although the door to balloon time has been significantly decreased, overall in-hospital mortality has not significantly declined in patients with ST elevation myocardial infarction (STEMI) undergoing primary percutaneous coronary intervention (PCI) [1, 5]. To achieve safe and effective therapeutic benefits, reperfusion therapy should be performed within 12 h of symptom onset as the therapeutic window [6]. Paradoxically, timely myocardial reperfusion is the cornerstone of therapy for acute STEMI [7]. However, this process leads to myocardial injury and cardiomyocyte death, known as myocardial reperfusion injury, which disrupts the therapeutic effects of reperfusion [7, 8]. Currently, no effective therapies or potential therapeutic targets are available for preventing reperfusion injury in STEMI [7, 9]. Therefore, the application of active adjunctive therapy to extend the critical therapeutic window and prevent reperfusion injury may improve clinical outcomes in patients with AMI. As the extent of myocardial salvage is an important determinant of the final infarct size in AMI, attenuation of ischemic/reperfusion (I/R) injury is critical for novel therapeutic strategies [10].

Targeted temperature management (TTM, which involves therapy hypothermia (TH) or prophylactic controlled normothermia) has been widely used as a gold standard treatment for minimizing secondary brain damage and improving neurologic outcomes in survivors of sudden cardiac arrest [11–14]. Although mild therapeutic hypothermia of TTM at 32–34°C improves the survival and neurologic outcomes of patients who have been successfully resuscitated after cardiac arrest, a study comparing TTM at 33°C and 36°C after cardiac arrest showed that TTM at 33°C was not beneficial compared to TTM at 36°C in patients with out-of- hospital cardiac arrest of presumed cardiac aetiology [11, 12]. As a new concept regarding TTM, the 2015 American Heart Association Guidelines for Cardiopulmonary Resuscitation and Emergency Cardiovascular Care recommended selecting and maintaining a constant target temperature of 32–36°C over a duration of at least 24 h in patients with return of spontaneous circulation after cardiac arrest [14]. TTM may be a promising strategy for improving myocardial salvage and cardiac function [15]. Several studies showed that a core temperature of <35°C during reperfusion limits the infarct size. However, this goal core temperature is not always achieved [16]. Therapeutic hypothermia commonly induces harmful effects, including bradycardia, atrial and ventricular arrhythmia, decreased cardiac output, and mild diastolic dysfunction [17]. The optimal target temperature and duration are unknown in established post-cardiac arrest care [14]. Considering all of the expected benefits and disadvantages according to the target temperatures during TTM, determining the optimal target temperature that clinically improves the outcomes of patients with myocardial I/R injury remains challenging.

Although the pathophysiology of myocardial I/R injury is very complex and poorly understood, inflammatory response and apoptotic cell death are known to play an important role in the development of ischemic heart damage by myocardial I/R injury [18, 19]. Apoptosis is an important mechanism in I/R injury, and therapeutic hypothermia reduces apoptosis in myocytes. Therapeutic hypothermia-induced myocardial protection is significantly associated with beneficial modifications in apoptotic signal pathways [18]. High mobility group box-1 (HMGB1), which is involved in the structural organization of DNA in eukaryotic cells, serves

as a primary mediator of the innate immune response after release by necrotic cells or active release during sterile injury [20]. HMGB1 is rapidly released upon I/R injury and is elevated after 30 min of ischemia [21]. Extracellular HMGB1 binding to Toll-like receptor 4 enhances the inflammatory response to myocardial damage after I/R and induces cardiomyocyte apoptosis [19, 20]. Therefore, synergistic interactions between HMGB1 and inflammatory factors amplify inflammatory responses and increase damage after I/R injury [19]. Plasma levels of HMGB1 are independently associated with increased mortality of STEMI patients treated with PCI [22]. Intravenous administration of glycyrrhizin, which attenuates extracellular release of HMGB1, significantly reduces the infarct size and decreased the levels of serum HMGB1, tumour necrosis factor (TNF)-α, and interleukin (IL)-6 [23]. In a previous study using a middle cerebral artery occlusion rat model, both glycyrrhizin-mediated inhibition of HMGB1 and intracerebroventricular neutralizing antibody treatment significantly reduced the infarct volume [24]. Thus, HMGB1 is a valuable molecular target for new adjunctive therapies that extend the critical therapeutic window by blocking sterile inflammation during early myocardial damage after I/R injury. The exact mechanism by which hypothermia attenuates myocardial damage due to ischemia and reperfusion remains unknown [25]. It is critical to understand the direct functional and mechanistic relationships between TTM and HMGB1 in a clinically relevant model of AMI.

We previously demonstrated that TTM at both 33˚C and 36˚C equivalently helped rescue ischemic penumbra from exacerbated ischemic injury by attenuating pro-inflammatory cytokine production via HMGB1 blockade in a clinically relevant middle cerebral artery occlusion rat model [24, 26]. Although TTM at 36˚C is advantageous for ameliorating hypothermia-induced cardiac arrhythmia, shivering, and rewarming damage, whether core temperatures of 36˚C and 33˚C are equally effective in our preclinical model of left anterior descending (LAD) coronary artery ligation remains unclear.

We hypothesized that TTM could attenuate the inflammatory response in peri-infarct regions by inhibiting the extracellular release of HMGB1 using a rat LAD coronary artery ligation model and subsequently reduce the myocardial infarcted area, resulting in increased myocardial protection after I/R injury. We investigated whether TTM at 36˚C has a myocardial protective effect via the same mechanism.

## Methods

### Preparation of experimental animals

Healthy, age-matched, adult male Wistar rats weighing 400–430 g were acquired from a single source breeder at Orientbio (Seongnam, Republic of Korea). All experiments and animal care were conducted in strict accordance with guidelines and protocols approved by the Institutional Animal Care and Use Committee of the Yonsei University Health System (2016–0043) and National Institutes of Health.

### Experimental rat model of myocardial I/R injury

Before surgery of the experimental rat model, anaesthesia was induced with 5% isoflurane in a mixture of 0.7 L/min nitrous oxide and 0.3 L/min oxygen and maintained using 2% isoflurane in the same gas mixture. After anaesthesia, tracheostomy was conducted using a midline neck incision and intravenous catheter (4712-020-116. I.V Catheter 16G, Sewoon Medical Co., Cheon-An, Korea). Mechanical ventilation (tidal volume, 3.0 mL; respiratory rate, 50/min) was supported by a rodent ventilator (SAP-830/AP, CWE, Inc., Ardmore, PA). The heart was exposed by left vertical thoracotomy and pericardiectomy. Ligation of the LAD coronary artery was performed on rats as described previously [27]. An LAD coronary artery

was ligated at the mid portion between the pulmonary artery and apex through a 6–0 ethilon suture. Immediately before ligation, the PE-10 tube (polyethylene tube, OD 0.61 mm) was placed between the LAD and suture. The suture was ligated with the PE-10 tube. Ischemia was confirmed, with cyanosis and dyskinesia of the myocardium supplied by LAD observed to be developed after ligation. Reperfusion was induced by removing the PE-10 tube after 30 min of LAD ligation and was sustained for 3 h 30 min. The skin was closed with 4–0 nylon sutures after reperfusion. The same surgical procedures were performed in sham animals except for ligation [27]. After 4 hours of LAD ligation, anaesthesia was performed with 5% isoflurane in a mixture of 0.7 L / min nitrous oxide and 0.3 L / min oxygen by inhalation. and euthanasia was carried out.

## Experiment protocol

We divided the present study into two main experiments. To assess the effects of myocardial protection exerted by TTM at 33˚C and 36˚C, the rats were randomly divided into four experimental groups: sham + 37˚C (n = 5), sham + 33˚C TTM (n = 5) sham + 36˚C TTM (n = 5), LAD I/R + 37˚C normothermia (n = 5), LAD I/R + 33˚C TTM (n = 5), and LAD I/R + 36˚C TTM (n = 5). The target core temperature was monitored in the rectum of rats and maintained during all experiments using a feedback-controlled heating pad (HB 101, Harvard Apparatus, Holliston, MA, USA). In the sham and normothermic groups, the target core temperature temperatures were maintained at 37.0 ± 0.5˚C. In the TTM groups with target temperatures of 33˚C and 36˚C, external surface cooling was started at 15 min after LAD coronary ligation by placing ice packs on the animal's torso. The TTM target temperatures of 33˚C and 36˚C were maintained at 33.0 ± 0.5˚C and 36.0± 0.5˚C, respectively. To prevent shivering caused by TTM, vecuronium (0.9 mg/kg) was injected intramuscularly into all animals. Glycyrrhizin is a pharmacological inhibitor of HMGB1 and has been suggested to prevent HMGB1 release from cells by directly binding to HMGB1 [27–29]. To verify the cardioprotective effect of TTM by specifically inhibiting HMGB1 in our animal model, rats were randomly assigned to three different experimental groups: sham (n = 5), LAD I/R (n = 5), and LAD I/R after pre-treatment with glycyrrhizin (n = 5). Glycyrrhizin (100 mg/kg) was injected intraperitoneally into the rats at 30 min before the ligation of the LAD coronary artery.

## Assessment of infarct volume

To assess myocardial infarction, 2,3,5-triphenyltetrazolium chloride (TTC) (T8877, Sigma-Aldrich, St. Louis, MO, USA) staining was performed. The chest of anesthetized rats was re-opened at 4 h after sham treatment or LAD I/R surgery. The heart was quickly removed and sectioned into 2-mm-thick slices in a pre-chilled coronal matrix device (HSRA001-1, Zivic Instruments, Pittsburgh, PA, USA). Coronal sections were immersed for 30 min in a 1% TTC solution in sterile distilled water at 37˚C and then fixed in 4% paraformaldehyde in phosphate-buffered saline for 48 h. Each stained section was scanned with a flatbed scanner (PERFECTION V800 PHOTO, EPSN, Nagano, Japan). To measure the infarct volume, heart tissue between 0 and 8 mm from the apex of the heart was used. We measured the infarcted area in the anterior and posterior sides of each 2-mm-thick slice using ImageJ 1.48v software. To determine the infarct volume in each slice, the average value of the infarct area on the anterior and posterior sides was multiplied by the thickness (2mm) [thickness × (top area + bottom area)/2]. In addition, the total infarct volume was calculated as the sum of the infarct volume per slice.

## Immunohistochemistry analysis

For immunohistochemistry analysis, 2,3,5-TTC staining was performed to confirm the peri-infarct area in the left ventricle [28]. Next, 2-mm-thick slices between 4 and 6 mm from the apex of the rat heart were selected, fixed with a 4% paraformaldehyde solution and embedded in paraffin. Between 4 and 6 mm from the apex of the rat heart was chosen because the peri-infarcted region was easily observable given that it was properly mixed with normal and infarct tissue after TTC staining. Using a microtome (LEICA RM 2335, Wetzlar, Germany), the heart sections were cut at 4 μm thickness on New Silane III-coated microslides (Muto Pure Chemical, Tokyo, Japan) from a region including the infarct area. The sections were permeabilized and blocked with citrate buffer, 3% $H_2O_2$, and 5% bovine serum albumin in Tris-buffered saline (TBS) for 1 h at room temperature (RT). The sections were incubated in TBS containing Tween 20 and anti-HMGB1 polyclonal primary antibody overnight at 4˚C (1:100, ab18256; Abcam, Cambridge, UK). The sections were washed three times with TBS for 5 min and incubated for 1 h at RT with fluorescent secondary antibodies conjugated to Alexa-fluor 594 (1:100, A11032; Invitrogen, Carlsbad, CA, USA). The sections were washed three times with TBS and mounted with ProLong™Diamond Antifade Mountant containing DAPI (P36962, Invitrogen). The peri-ischemic areas of stained sections were observed with a confocal microscope (LSM 700; Carl Zeiss GmbH, Jena, Germany).

## Enzyme-linked immunosorbent assay (ELISA) for cardiac troponin T (cTnT) and HMGB1

To obtain serum samples from rats, blood was drawn from the right atrium at 4 h after ligation of the LAD coronary artery with a 22-gauge needle. One millilitre of collected blood was transferred into a Z Serum Sep Clot Activator (Greiner Bioone, Kremsmunster, Austria), followed by centrifugation for 15 min at 3,000 rpm. The cTnT concentrations were determined using an cTnT ELISA kit (MBS2024997, MyBioSource, San Diego, CA, USA) and and HMGB1 concentrations were determined using the Rat HMGB1 ELISA kit (Solarbio, Beijing, China).

## Real-time polymerase chain reaction (RT-PCR)

To prepare peri-infarcted myocardium tissue, 2,3,5-TTC staining was conducted to confirm the peri-infarct area in the left ventricle [29]. Tissue RNA was isolated using a Hybrid-R kit (305–010, GeneAll Biotechnology, Seoul, Korea). PrimerQuest (IDT, Skokie, IL, USA) was used to design primers for glyceraldehyde-3-phosphate dehydrogenase, TNF-α, IL-1β, and IL-6. Single-stranded cDNA was synthesized from 500 ng of total RNA using the PrimeScript 1st strand cDNA Synthesis Kit (6110A, Takara Bio, Shiga, Japan) (S1 Table). Quantitative PCR was performed using a 7500 ABI system (Applied Biosystems, Foster City, CA, USA) utilizing the SYBR-Green reagent (Q5602, Gendepot, Katy, TX, USA).

## TUNEL assay

Apoptotic cells were detected by terminal deoxynucleotidyl transferase (TdT)-mediated dUTP nick end labeling (TUNEL) using DeadEndTM Fluorometric TUNEL system (Promega, WI, USA) according to the manufacturer's instructions. A confocal microscope (LSM700, Carl Zeiss GmbG, Jena, Germany) was used to identify the stained sections. One slide from each animal was selected and stained. The two peri-ischemic areas of the stained sections were observed with a confocal microscope (LSM 700; Carl Zeiss GmbH, Jena, Germany). The average values of TUNEL-positive cells in the peri-infarct area were derived from two areas on the

stained sections. Numbers of TUNEL-positive cells in the infarct area were normalised using the numbers from the hearts of sham animals.

## Statistical analysis

All experimental results are expressed as the mean ± standard deviation of the mean. Statistical analyses were performed using unpaired $t$-test or by one-way analysis of variance (ANOVA) followed by Bonferroni *post hoc* tests for multiple comparisons between groups. Differences with $P < 0.05$ were considered as significant.

## Results

### Target temperatures of 33˚C and 36˚C equivalently reduce infarct volume in myocardial I/R injury

The core target temperatures of 33˚C ± 0.5˚C and 36˚C ± 0.5˚C were reached within 13 ± 0.80 and 5 ± 0.49 min after the onset of TTM. In the present study, the average values of the core temperature on reperfusion were 33.2˚C ± 0.07˚C in the 33˚C group and 35.8˚C ± 0.05˚C in the 36˚C groups (Fig 1A and 1B). Infarct volumes were assessed by TTC staining after 4 h of ischemic injury (Fig 1C and 1D). In the normothermic group, the mean ratio of the infarcted area after myocardial I/R was 15.7 ± 3.55% compared to the total area between 0 and 8 mm from the apex, whereas the mean ratio of the infarcted area at 33˚C and 36˚C of TTM was 6.9 ± 1.66% and 6.28 ± 3.05%, respectively. There was a significant difference between the normothermic group and TTM groups ($P = 0.001$).

To detect differences in myocardial protective effects at 33˚C and 36˚C in TTM after myocardial I/R injury, we compared the degree of reduction of the infarct volume in the 33˚C and 36˚C TTM groups. There was no significant difference between the infarct volumes of the 33˚C TTM group and the 36˚C TTM group ($P = 0.999$). These results suggest that application of TTM at both 33˚C and 36˚C has significant myocardial protective effects and that these temperatures lead to equivalent protection against myocardial I/R injury.

### Different target temperatures of 33˚C and 36˚C TTM similarly suppress extracellular release of HMGB1 from peri-infarct tissue after myocardial I/R injury

When ischemic damage to the myocardium is induced by LAD ligation of the heart, HMGB1 is released from the nucleus of myocardial cells [30, 31].

We found that the HMGB1 immunoreactivity was significantly decreased in the myocardium after ligation of LAD in rats. To investigate whether TTM at 33˚C and 36˚C significantly reduced the release of extracellular HMGB1 following I/R injury, we compared HMGB1 immunoreactivity between the normothermia and TTM groups after LAD ligation. We found that 21.15 ± 7.29% of 4,6-diamidino-2-phenylindole (DAPI)-positive cells in the peri-ischemic myocardium of LAD ligation rats were HMGB1-positive. However, we also found that target temperatures of 33˚C and 36˚C similarly restored the number of HMGB1-positive cells in post-infarct tissues. The percentages of HMGB1-positive cells were 81.28 ± 5.21% and 76.68 ± 6.27% for TTM at 33˚C and 36˚C, respectively. While significant increases for the proportion of HMGB1-positive cells were observed for TTM at 33˚C and 36˚C compared to the normothermic group ($P < 0.001$), there was no significant difference between the 33˚C and 36˚C groups ($P = 0.999$). This suggests that both 33˚C and 36˚C TTM cause similarly significant reductions in the extracellular release of HMGB1 after ischemic myocardial damage (Fig 2A and 2B).

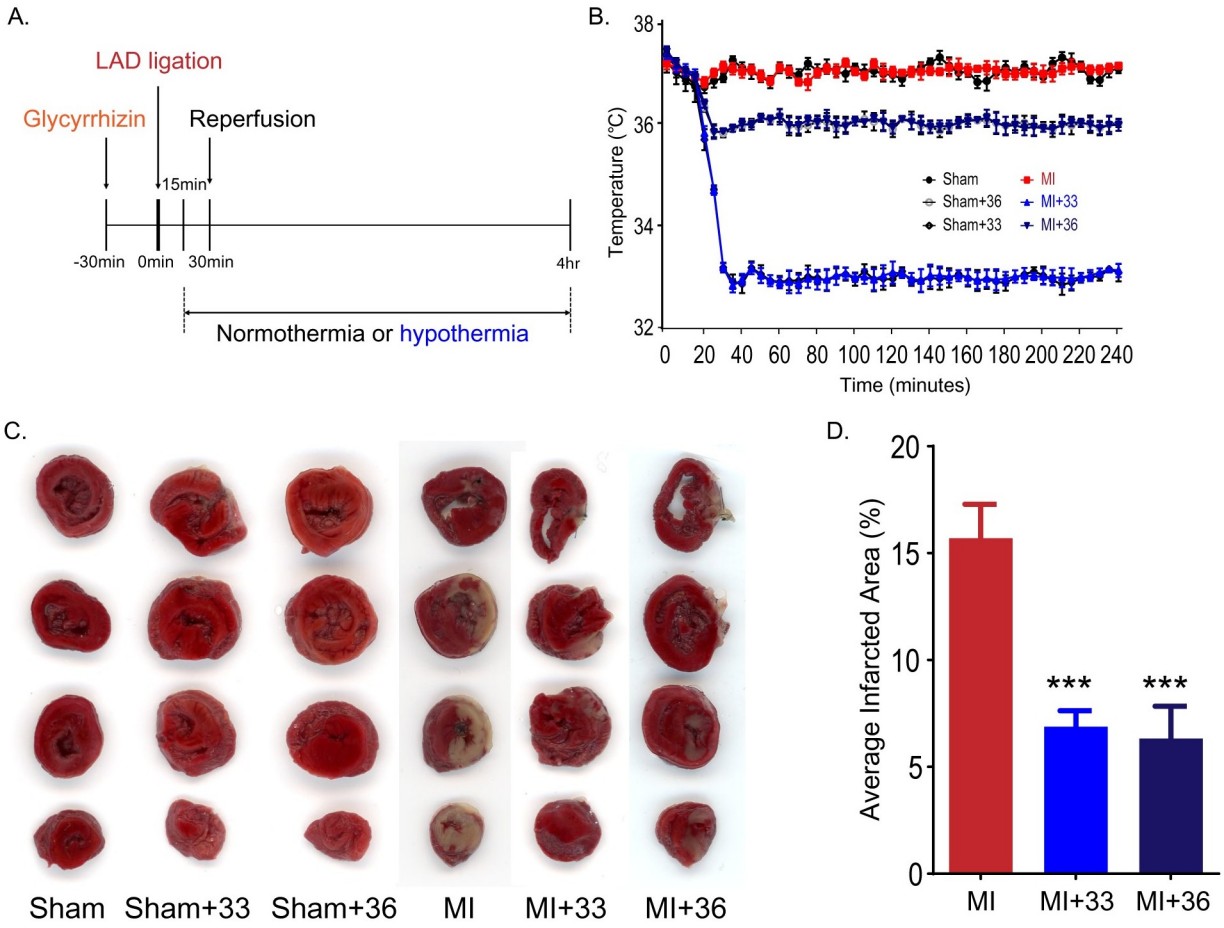

**Fig 1. Targeted temperature management at 33˚C and 36˚C similarly reduces infarct volume in myocardial I/R injury.** A. Experimental schedule. B. Changes in rat body temperature after LAD ligation (the number of animals: n = 5, respectively). C. Representative image of 2,3,5-triphenyltetrazolium chloride (TTC) staining. D. Volume of myocardial infarction stained with TTC (the number of animals: n = 5, respectively). $***P < 0.001$, comparison of myocardial I/R with normothermia and hypothermia (33˚C and 36˚C), Statistical analyses were performed by one-way analysis of variance (ANOVA) followed by Bonferroni *post hoc* tests for multiple comparisons between groups.

## TTM at 33˚C and 36˚C similarly inhibited inflammatory cytokine expression from peri-infarct regions

Cardiac mRNA expression of three major inflammatory cytokines (i.e., TNF-α, IL-1β, and IL-6) was assessed by quantitative RT-PCR in the peri-infarcted myocardium 4 h after LAD ligation. In normothermic rats maintained at 37˚C after myocardial I/R, the expression levels of TNF-α ($3.28 \pm 1.62$, $P = 0.001$), IL-1β ($36.15 \pm 18.5$, $P < 0.001$), and IL-6 ($1055.89 \pm 185.63$, $P < 0.001$) were significantly increased. Compared to the normothermic I/R group, TTM treatment at 33˚C was closely associated with lower expression of inflammatory cytokines in the peri-infarcted myocardium (TNF-α ($0.67 \pm 0.23$, $P = 0.001$), IL-1β ($2.67 \pm 1.33$, $P < 0.001$), IL-6 ($98.43 \pm 42.12$, $P < 0.001$) (Fig 2C, 2D, and 2E)). TTM at 36˚C also decreased the expression of these cytokines in the peri-infarcted myocardium (TNF-α ($0.67 \pm 0.19$, $P = 0.001$), IL-1β ($3.49 \pm 1.31$, $P < 0.001$), IL-6 ($98.68 \pm 50.89$, $P < 0.001$)). First, there were no significant differences in the mRNA expression of three inflammatory cytokines between the 33˚C and 36˚C TTM groups ($P < 0.999$, $P = 0.999$, $P < 0.999$, respectively). Thus, the application of TTM prevents the aggravation of damage by suppressing the production of inflammatory cytokines in

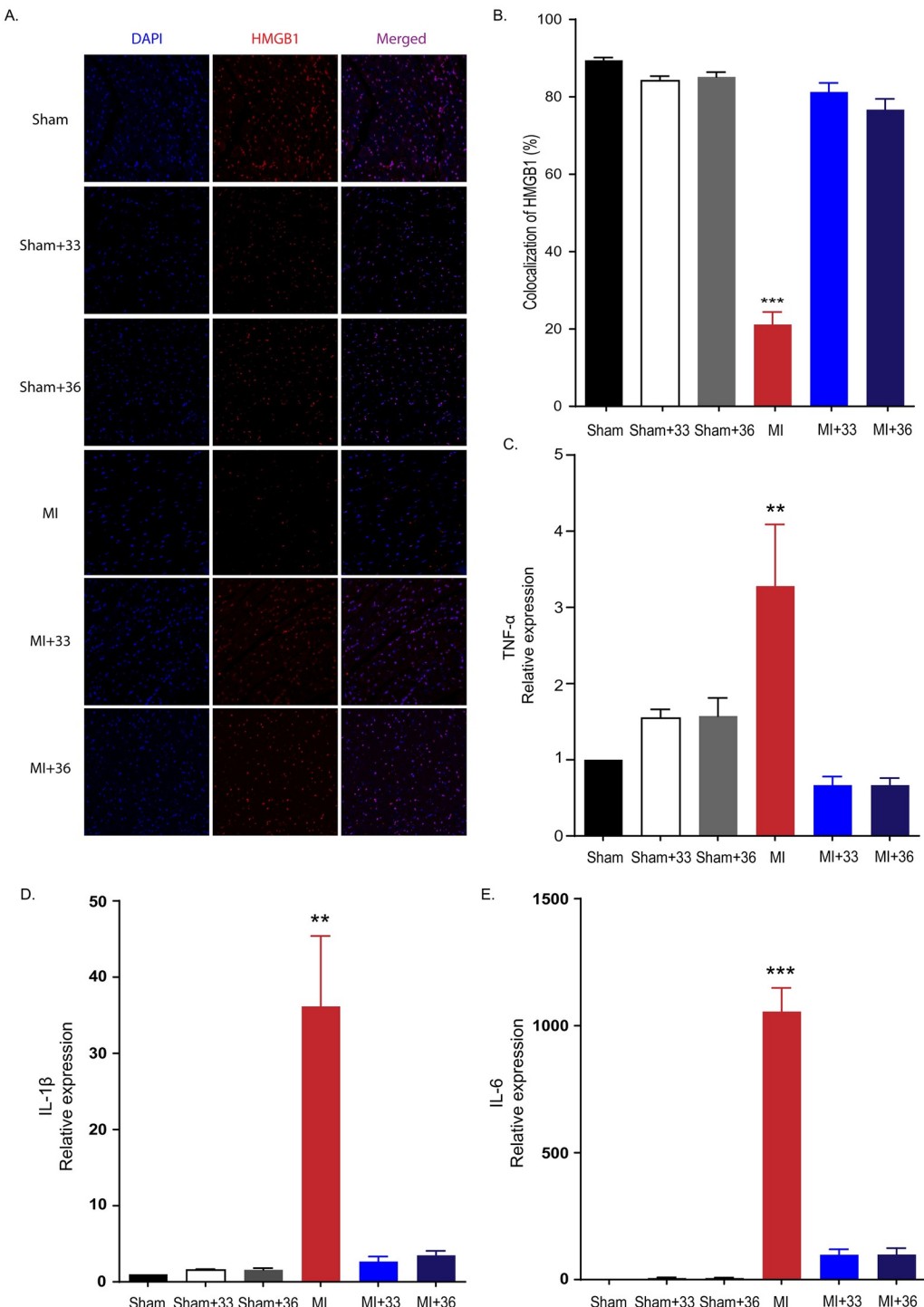

**Fig 2. Hypothermia suppresses extracellular release of HMGB1 after myocardial I/R injury and inflammatory cytokine expression in peri-infarct regions.** A. Representative immunohistochemistry results for 33˚C and 36˚C targeted temperature management after myocardial I/R injury. B. Immunohistochemistry results (the number of animals: n = 5, respectively), $^{***}P < 0.001$, comparing myocardial I/R with normothermia and hypothermia (33˚C and 36˚C), one-way analysis of variance (ANOVA), followed by Bonferroni *post hoc* test. C. Quantification of tumour necrosis factor-α (TNF-α) expression by RT-PCR (the number of animals: n = 5, respectively), $^{**}P < 0.01$, comparison of myocardial I/R with normothermia and hypothermia (33˚C and 36˚C) by one-way analysis of variance (ANOVA) followed by Bonferroni *post hoc* test. D. Quantification of interleukin-1β (IL-1β) expression by RT-PCR (the number of animals: n = 5, respectively), $^{**}P < 0.01$. Comparison of myocardial I/R with normothermia and hypothermia (33˚C and

36˚C) by ANOVA followed by Bonferroni *post hoc* test. E. Quantification of IL-6 expression by RT-PCR (the number of animals: n = 5, respectively), ***$P < 0.01$. Comparison of myocardial I/R with normothermia and hypothermia (33˚C and 36˚C) by ANOVA followed by Bonferroni *post hoc* test. All statistical analyses were performed by one-way analysis of variance (ANOVA) followed by Bonferroni *post hoc* tests for multiple comparisons between groups.

the peri-infarct area after myocardial I/R injury (Fig 2C, 2D, and 2E). This indicates that the different target temperatures of 33˚C and 36˚C TTM similarly attenuate inflammatory cytokine expression after cardiac I/R injury.

### TUNEL assay

After myocardial I/R, TUNEL-positive apoptotic cells, which appear as light green dots under the confocal microscope, were significantly increased in the normothermic group (81.7 ± 16.11) compared to the 33˚C (19.4 ± 7.19; $P < 0.001$) and 36˚C (13.7 ± 5.12; $P < 0.001$) TTM groups. However, there was no significant difference between the number of TUNEL-positive cells of the two TTM groups ($P = 0.676$) (Fig 3). These results also imply that application of TTM at both temperatures has significant myocardial protective effects by reducing apoptosis and that these core temperatures lead to equivalent protection against myocardial I/R injury.

### Glycyrrhizin alleviates myocardial damage by suppressing the extracellular release of HMGB1 in myocardial I/R injury

Glycyrrhizin is a pharmacological HMGB1 inhibitor that binds directly to HMGB1 and prevents the extracellular release of HMGB1 to block its cytokine function [32–34]. We compared the effects of glycyrrhizin treatment on infarct volume, extracellular release of HMGB1, expression of inflammatory cytokines, and plasma level of cTnT in our animal model. In the myocardial I/R group treated with an intra-peritoneal injection of glycyrrhizin, the infarct volume was significantly decreased (7.5 ± 3.81%) compared to that in the normothermic myocardial I/R group (15.3 ± 5.17%, $P = 0.04$) ((Fig 4A and 4B). Glycyrrhizin also significantly increased the proportion of HMGB1-positive cells in the I/R injured myocardium (21.52 ± 3.94% in normothermic rats after myocardial I/R versus 86.61 ± 3.65% in glycyrrhizin-treated myocardial I/R rats, $P < 0.001$) (Fig 4C and 4D). In glycyrrhizin–treated AMI rats, TNF-α (2.18 ± 0.77, $P < 0.001$), IL-1β (4.59 ± 0.95, $P < 0.001$), and IL-6 (145.78 ± 107.26, $P < 0.001$) levels were decreased compared to those in the myocardial I/R group (TNF-α; 6.18 ± 3.42, IL-1β; 63.78 ± 25.36, and IL-6; 2565.74 ± 707.31, respectively) (Fig 5A, 5B, and 5C). Additionally, cTnT levels were significantly lower in the glycyrrhizin-treated myocardial I/R group (0.80 ± 0.12 ng/mL) than in the normothermic group after myocardial I/R (2.39 ± 0.83 ng/mL, $P = 0.001$) (Fig 5E).

### Effects of 33˚C and 36˚C TTM on cTnT and HMGB1 levels in the plasma

To examine the myocardial protective effects of 33˚C and 36˚C TTM on cTnT levels reflecting myocardial damage, we measured cTnT levels in the plasma. The levels of cTnT were higher in the normothermia group after LAD ligation compared to those in the sham-operated group (1.75 ± 0.53 and 0.10 ± 0.05 ng/mL, respectively; $P < 0.001$). However, rats subjected to either 33˚C or 36˚C TTM showed lower cTnT levels than those in the normothermia group (0.33 ± 0.08 in 33˚C and 0.19 ± 0.07 ng/mL in 36˚C TTM group). There was no significant difference in plasma cTnT at 33˚C and 36˚C TTM ($P = 0.999$), indicating that both target core temperatures for TTM equivalently reduced myocardial damage (Fig 5D) Next, we performed an ELISA to measure HMGB1 levels in serum samples obtained at 4 h after the onset of

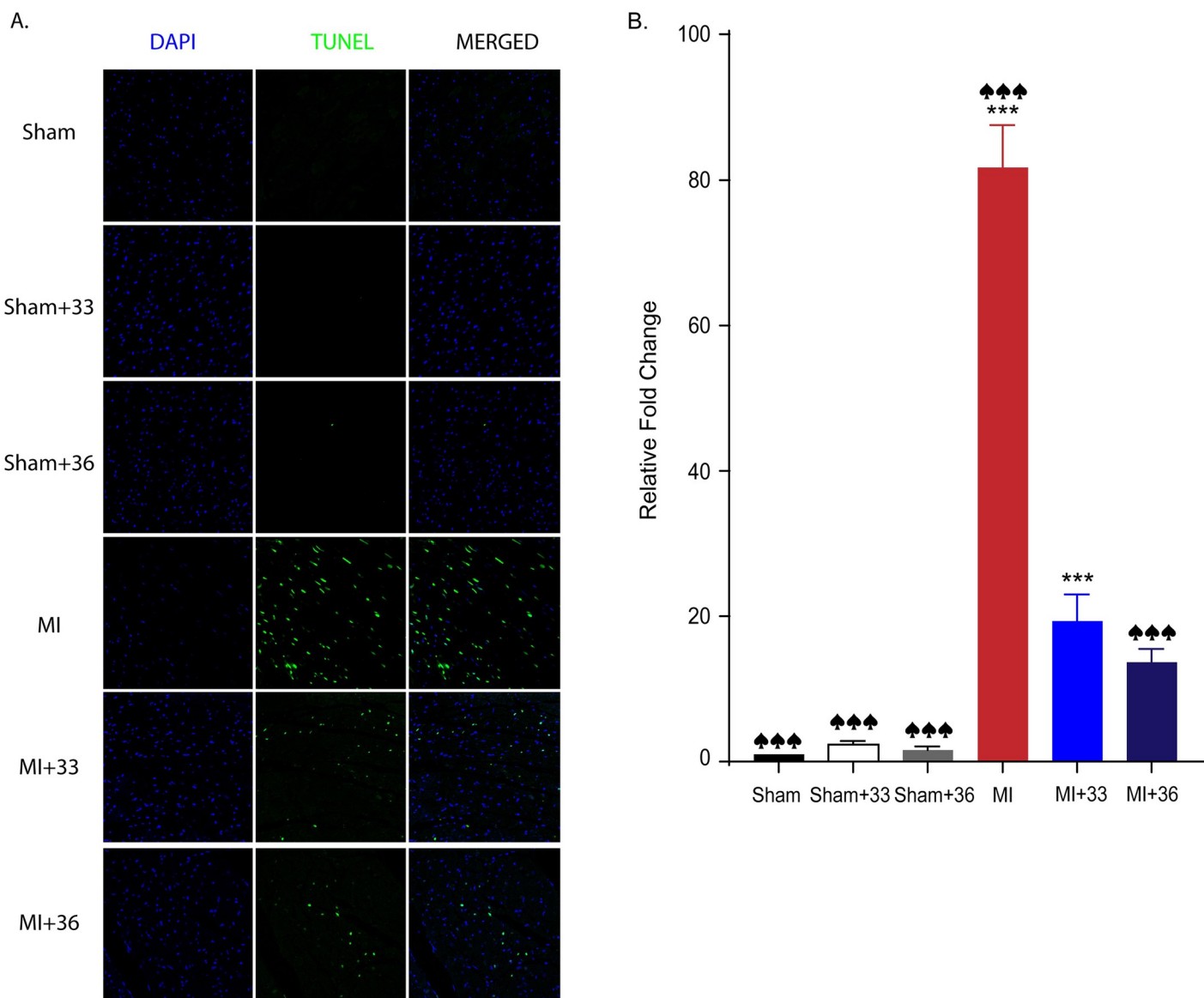

**Fig 3. Quantitative analysis of apoptotic cell death by TUNEL assay.** A. Representative TUNEL assay results for 33°C and 36°C targeted temperature management after myocardial I/R injury. B. TUNEL assay results (the number of animals: n = 5, respectively), ♠♠♠ $P < 0.001$, comparing sham with normothermia and hypothermia (33°C and 36°C), ***$P < 0.001$, comparing myocardial I/R with normothermia and hypothermia (33°C and 36°C); Statistical analyses were performed by one-way analysis of variance (ANOVA) followed by Bonferroni *post hoc* tests for multiple comparisons between groups.

ischemia. As expected, the level of circulating of HMGB1 was increased after I/R injury, but this increase was significantly attenuated by TTM at 33°C and 36°C (normothermic group after myocardial I/R, 367.08 ± 83.58 pg/mL, 33°C TTM after myocardial I/R, 67.15 ± 15.55 pg/mL and 36°C TTM after myocardial I/R, 66.30 ± 7.43 pg/mL, $P < 0.001$) (Fig 6).

## Discussion

Our results suggest that TTM at both 33°C and 36°C reduces myocardial injury following acute myocardial I/R injury by suppressing the extracellular release of HMGB1. We found that

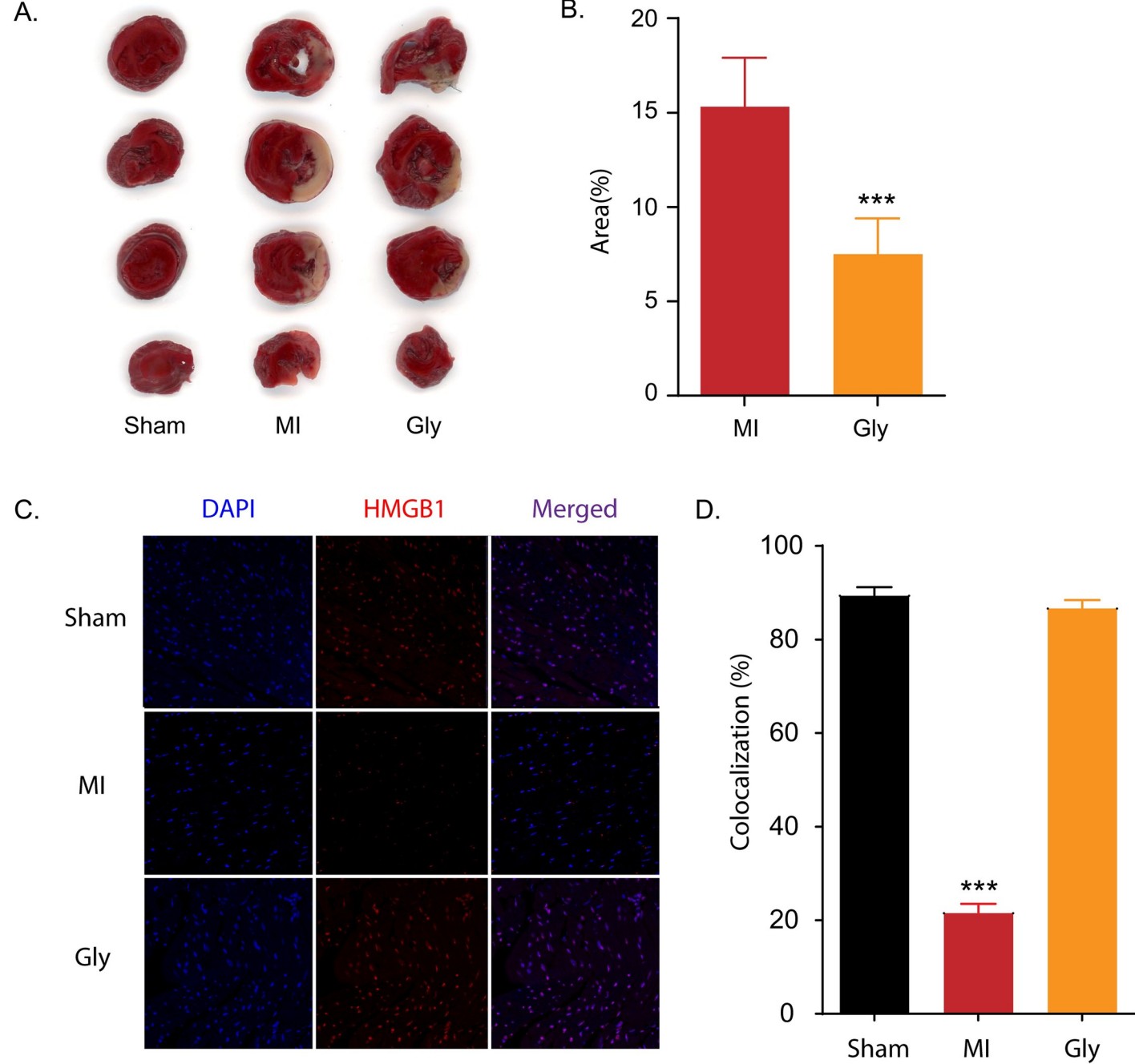

**Fig 4. Glycyrrhizin reduces infarct volume in myocardial I/R injury and glycyrrhizin suppresses extracellular release of HMGB1 in myocardial I/R injury.** To verify the cardioprotective effect of TTM by specifically inhibiting HMGB1, rats were assigned to sham, LAD I/R, and LAD I/R after pre-treatment with glycyrrhizin (known as a pharmacological inhibitor of HMGB1) groups. A. Representative image for 2,3,5-triphenyltetrazolium chloride (TTC) staining comparing myocardial I/R injury with normothermia and glycyrrhizin pre-treatment. B. Quantification of TTC staining results in A (the number of animals: n = 5, respectively). ***$P < 0.001$ comparing myocardial I/R injury with and without glycyrrhizin, unpaired *t*-test. C. Quantification of immunohistochemistry results in D (the number of animals: n = 5, respectively). D. Representative images showing HMGB1 immunoreactivity from myocardial I/R injury with and without glycyrrhizin B treatment. ***$P < 0.001$, comparing myocardial I/R injury with and without glycyrrhizin by one-way analysis of variance (ANOVA) followed by Bonferroni *post hoc* test.

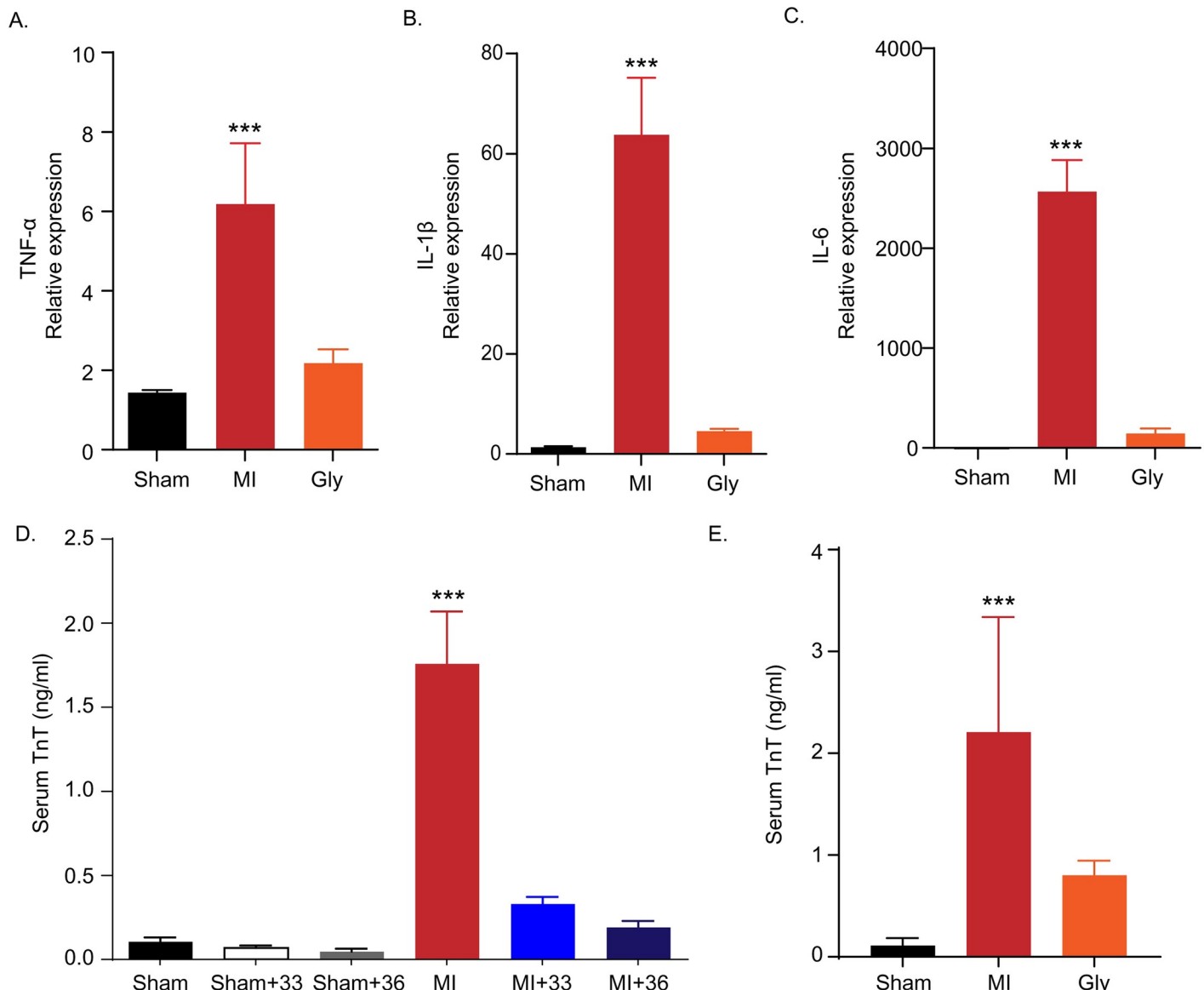

**Fig 5. Inflammatory cytokine expression in myocardial I/R injury with and without glycyrrhizin treatment and levels of cardiac troponin T (cTnT) in plasma.**
A. Quantification of tumour necrosis factor-α (TNF-α) expression by RT-PCR comparing myocardial I/R injury with and without glycyrrhizin treatment (the number of animals: n = 5, respectively), ***P < 0.001, comparing myocardial I/R injury with and without glycyrrhizin by one-way analysis of variance (ANOVA) followed by Bonferroni *post hoc* test B. Quantification of interleukin-1β (IL-1β) expression by RT-PCR comparing myocardial I/R injury with and without glycyrrhizin (the number of animals: n = 5, respectively), ***P < 0.001, comparing myocardial I/R injury with and without glycyrrhizin treatment by one-way ANOVA followed by Bonferroni *post hoc* test C. Quantification of interleukin-6 (IL-6) expression by RT-PCR comparing myocardial I/R injury with and without glycyrrhizin treatment (the number of animals: n = 5, respectively), ***P < 0.001, comparing myocardial I/R injury with and without glycyrrhizin treatment by one-way ANOVA followed by Bonferroni *post hoc* test. D. Quantification of serum TnT level by ELISA comparing myocardial I/R with normothermia and hypothermia (33˚C and 36˚C) (number of animals: n = 5), ***P < 0.001, comparison of myocardial I/R with normothermia and hypothermia (33˚C and 36˚C) by one-way analysis of variance (ANOVA) followed by Bonferroni *post hoc* test. E. Quantification of serum TnT level by ELISA with comparing myocardial I/R injury with and without glycyrrhizin pre-treatment (number of animals: n = 5), ***P < 0.001, comparing myocardial I/R injury with and without glycyrrhizin pre-treatment by one-way ANOVA followed by Bonferroni *post hoc* test.

TTM attenuated the increase in infarct size, apoptosis, extracellular release of HMGB1, and pro-inflammatory effect against the propagation of injury in rats with AMI. We also showed that TTM at both 33˚C and 36˚C significantly attenuated the elevation of cardiac troponin, which is a sensitive and specific marker of heart muscle damage, after myocardial I/R injury.

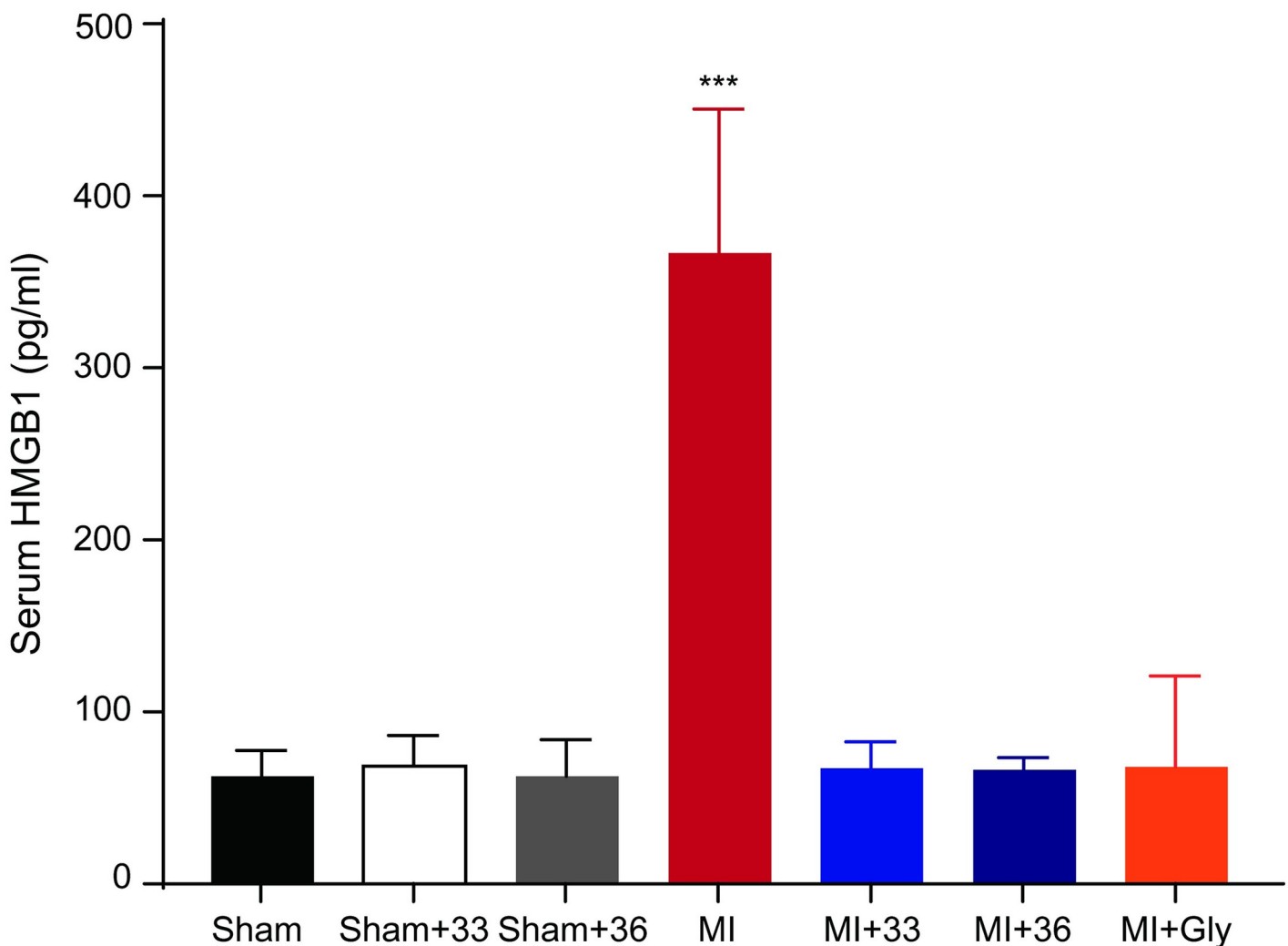

**Fig 6. Quantification of levels of HMGB1 in plasma.** Quantification of serum HMGB1 level by ELISA comparing myocardial I/R with normothermia, hypothermia (33°C and 36°C), and glycyrrhizin pre-treatment (number of animals: n = 5), ***P < 0.001, Statistical analyses were performed by one-way analysis of variance (ANOVA) followed by Bonferroni *post hoc* tests for multiple comparisons between groups.

TTM at 36°C showed similar myocardial protective effects against myocardial I/R injury as TTM at 33°C in our rat AMI model.

The inflammatory cascade in myocardial injury and infarction is significantly associated with debris removal and scar formation [9]. Despite the fundamental role of inflammation in wound healing after AMI, an overwhelming inflammatory response can lead to devastating effects on cardiomyocytes [9]. The onset of cell death begins within 30 min to 1 h after the cessation of blood flow through a combination of necrosis and apoptosis. Inflammation, which contributes to lethal myocardial injury, is initiated during ischemia and is sustained over several hours after reperfusion [35]. Inhibiting the inflammatory process can provide a potential therapeutic window for cardioprotection [35]. HMGB1 is known to subsequently act as a pro-inflammatory cytokine that activates inflammatory cells by its extracellular release from necrotic cells in the pathophysiology of various diseases [20, 36, 37]. The serum concentration of HMGB1 is significantly associated with infarct size and residual left ventricular function in patients with STEMI [38]. Furthermore, extracellular HMGB1 mediates inflammation and

enhances the regeneration of damaged tissues. Takahashi et al. reported that HMGB1 has beneficial effects at low concentrations and deleterious effects at high concentrations [36]. HMGB1 levels are significantly increased within 1 h and maintained for up to 24 h during I/R injury [39, 40]. Therefore, it is important to prevent the action of HMGB1 to alleviate ischemic injury of the myocardium [20, 41]. Nevertheless, HMGB1 plays dual roles in cardiac injury. In the initial stage of cerebrovascular and cardiovascular diseases, HMGB1 is released from the cell to participate in the cascade amplification reaction of inflammation, causing vasospasm and apoptosis [20, 41, 42]. In the recovery stage of disease, HMGB1 can promote tissue repair, regeneration, and remodelling [20, 41, 42]. It is necessary to investigate whether the newly generated HMGB1 plays a role in propagating inflammatory detriment or repairing damages over time after injury. The interaction between extracellular HMGB1 and Toll-like receptor 4 enhances the inflammatory response to myocardial damage after I/R by activating the release of pro-inflammatory cytokines, such as TNF-$\alpha$ from macrophage/monocytes [20]. TNF-$\alpha$ influences the production of other pro-inflammatory cytokines, such as IL-1 $\beta$ and IL-6, resulting in a negative cycle of pro-inflammatory cytokine production and aggravation of injury after myocardial infarction [43, 44]. We also demonstrated that TTM attenuated myocardial I/R-induced apoptosis. Temperatures of 33°C or 36°C in TTM induced equivalent myocardial protection by attenuating apoptosis after I/R injury.

Our previous study demonstrated that therapeutic application of TTM helps alleviate peri-infarct damage from the propagation of ischemic injury in an ischemic stroke model by reducing inflammatory cytokines through the blockage of HMGB1 release [24, 26]. This is the first study to show a direct mechanistic and functional link between HMGB1 and TTM in a clinically relevant AMI animal model. Interestingly, myocardial I/R injured rats treated with glycyrrhizin showed equivalent myocardial protection as myocardial I/R rats that underwent TTM. This suggests that the extracellular release of HMGB1 is critical for the propagation of I/R injury by increasing the expression of proinflammatory cytokines in the peri-infarct myocardium and that TTM helps attenuate this propagation of I/R injury by inhibiting HMGB1 after I/R injury. In addition, we demonstrated that the serum HMGB1 level was lower in the TTM group than in the group without TTM, revealing a correlation between TTM and HMGB1. However, there is a lack of data regarding the link between HMGB1 and I/R injury and direct mechanisms in our study. In the case of the AMI animal model, we could not confirm the relationship between HMGB1 and I/R injury because HMGB1 neutralizing antibodies could not be injected locally. Instead, we indirectly revealed the association of HMGB1 with I/R injury by using glycyrrhizin as a pharmacological HMGB1 inhibitor and via experiments using our middle cerebral artery occlusion model [24, 26]. However, these analyses with glycyrrhizin and neutralizing antibodies only demonstrated that the extracellular release of HMGB1 is a key factor in the cardiac damage after I/R injury. There is no evidence that the blockade of the HMGB1 release is direct mechanism of the TTM-induced cardiac protection after I/R injury. Further studies are needed to identify the direct mechanisms underlying, and the link between, HMGB1 release and the action of TTM after I/R injury. In addition, Liu et al. identified several classes of agents that potently induce the nucleo-cytoplasmic relocation and subsequent cellular release of HMGB1 [45]. To clarify whether TTM can attenuate the inflammatory response in peri-infarct regions by inhibiting the extracellular release of HMGB1 after I/R injury, further studies are needed to verify ischemic damage using pharmacological agents that induce HMGB1 release during TTM after I/R injury in the AMI model.

Induced hypothermia may increase the rates of lethal arrhythmia, hypotension, shivering, infection, impaired coagulopathy, and rewarming injury and significantly alter the pharmacokinetics [46]. Its intrinsic adverse effects can significantly diminish the hypothermic benefits throughout the body [46]. TTM at 36°C may be preferred to TTM at 33°C in patients with

cardiac arrest for several reasons [26]. In clinical practice, TTM consists of three phases: induction, maintenance, and rewarming [47]. TTM should be initiated as soon as possible according to international guidelines [46]. Moreover, rapidly induced hypothermia is important for modulating the efficacy of hypothermia in the clinical setting because minimizing the door to balloon time and reaching the target temperature within that at pre-reperfusion are critical for salvaging cardiac muscles [48]. As TTM at 36˚C is close to the lower margin of the normal body temperature, it has the advantage of quickly achieving the target core temperature [46]. Clinical management to control shivering and minimize the risk of the devastating complications of TTM should be considered to maximize the benefits [47]. Shivering, as a major adverse effect of TTM, leads to extremely uncomfortable and massive increases in the metabolic demand and systemic energy consumption [47]. TTM at 36˚C may be preferred to minimize the risk of shivering in the induction and rewarming phases because peripheral vasoconstriction and shivering are triggered at 36.5˚C and 35.5˚C in healthy humans [47]. In the rewarming phase, a small temperature change between the maintenance and rewarming periods can be beneficial for reducing the increased risk of secondary damage due to the adverse consequences of rewarming on the whole body. TTM at 33˚C is generally recommended as the safer margin for the target core temperature in critically ill patients because temperatures below 32˚C can induce serious cardiac arrhythmia. Application of TTM at 36˚C to AMI patients also helps ameliorate the risk of several adverse effects. Unlike patients who are resuscitated after cardiac arrest, most patients with AMI remain awake and breathe spontaneously during acute management [47]. Simple and well-tolerated TTM at 36˚C is more feasible during the acute period of AMI. In this experiment, we did not study the rewarming phase when applying 33˚C and 36˚C TTM. Rewarming treatment of the 33˚C TTM group is likely to be more damaging than 36˚C TTM group. However, due to the technical problems of our study, it was difficult to implement rewarming experiments. In future studies, additional rewarming treatments will be necessary because they can provide more meaningful interpretations compared to actual clinical practice.

TTM has been shown to be safe and feasible in clinical practice, and there were no differences in mortality or neurological outcomes between patients who underwent TTM at 33˚C and those who underwent TTM 36˚C after out-of-hospital cardiac arrest in recent multi-centre clinical trials [49]. Recently, the target temperature of TTM tends to change accordingly from 33–34˚C to 36˚C during post-resuscitation care [49]. However, previous animal studies demonstrated that rapid application of therapeutic hypothermia at 32–34˚C prior to reperfusion significantly reduced the myocardial infarct size [25, 50, 51]. Although the infarct size in TTM at 35˚C is decreased, Dash et al. demonstrated that TTM at 32˚C is superior to TTM at 35˚C and normothermic porcine after AMI [48]. Several clinical trials of TTM after AMI have shown inconsistent results with the major findings of many experimental studies.

This may be because of interspecies variability among animal models and differences in the immune response to I/R injury. Additionally, clinical outcomes may also be affected by disease- or organ-specific characteristics and molecular biological differences among organs [16, 17]. To reduce the gap between the TTM beneficial effects for cardiac arrest and AMI, we investigated whether TTM at 36˚C has a potent myocardial protective effect. Our study showed that target temperatures of 33˚C or 36˚C in TTM similarly inhibited HMGB1 release and induced equivalent myocardial protection in terms of the infarct size in myocardial I/R injury in a rat model. This is the first study to suggest that a target core temperature of 36˚C is applicable for cardioprotection in myocardial I/R injury. However, we compared the effects of two different temperatures in rats intubated with ventilation support, adequate sedation, and strict shivering control by vecuronium, mimicking the features of cardiac arrest. Therefore, further studies are needed to clarify the cardioprotective effects of both 33˚C and 36˚C TTM

and the critical roles of HMGB1 in patients with AMI who are awake and exhibit spontaneous breathing. Finally, the importance of this study is that whilst there are many described pathways known to protect against I/R injury due to TTM, here we identified a novel pathway that inhibits the release of HMGB1.

## Conclusion

We describe a new mechanistic and clinical link showing that TTM at 36˚C is a therapeutic candidate that should be investigated in future clinical trials by reducing the propagation of myocardial damage to effectively inhibit the extrahular HMGB1 release after myocardial I/R injury.

## Supporting information

**S1 Table. Sequences.**
(DOCX)

## Author Contributions

**Conceptualization:** Jin Ho Beom, Chul Hoon Kim, Je Sung You.

**Data curation:** Jin Ho Beom, Ju Hee Kim, Jeho Seo, Jung Ho Lee, Je Sung You.

**Formal analysis:** Jin Ho Beom, Yong Eun Chung, Chul Hoon Kim, Je Sung You.

**Funding acquisition:** Jin Ho Beom, Chul Hoon Kim, Je Sung You.

**Investigation:** Chul Hoon Kim, Je Sung You.

**Methodology:** Jin Ho Beom, Ju Hee Kim, Jeho Seo, Jung Ho Lee, Yong Eun Chung, Hyun Soo Chung, Sung Phil Chung, Chul Hoon Kim, Je Sung You.

**Project administration:** Chul Hoon Kim, Je Sung You.

**Supervision:** Chul Hoon Kim, Je Sung You.

**Visualization:** Jeho Seo.

**Writing – original draft:** Jin Ho Beom, Chul Hoon Kim, Je Sung You.

**Writing – review & editing:** Jin Ho Beom, Ju Hee Kim, Jeho Seo, Jung Ho Lee, Yong Eun Chung, Hyun Soo Chung, Sung Phil Chung, Chul Hoon Kim, Je Sung You.

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
