## [Decision Letter · Decision Letter 0]

22 Sep 2020

PONE-D-20-25658

Targeted temperature management by inhibiting HMGB1 release in myocardial ischemia/reperfusion injury

PLOS ONE

Dear Dr. Sung You,

Thank you for submitting your manuscript to PLOS ONE. After careful consideration, we feel that it has merit but does not fully meet PLOS ONE’s publication criteria as it currently stands. Therefore, we invite you to submit a revised version of the manuscript that addresses the points raised during the review process.

Specifically, the reviewers raised criticisms concerning the quality of the images, asked for more detailed methods and for the necessity to perform an ELISA in order to quantify HMGB1 plasma levels. Further, they also raised doubts on the statistical analysis performed.

We look forward to receiving your revised manuscript.

Kind regards,

Federica Limana

Academic Editor

PLOS ONE

Journal Requirements:

Reviewers' comments:

Reviewer's Responses to Questions

**Comments to the Author**

1. Is the manuscript technically sound, and do the data support the conclusions?

Reviewer #1: Partly

Reviewer #2: Yes

2. Has the statistical analysis been performed appropriately and rigorously? 

Reviewer #1: I Don't Know

Reviewer #2: Yes

3. Have the authors made all data underlying the findings in their manuscript fully available?

Reviewer #1: No

Reviewer #2: Yes

4. Is the manuscript presented in an intelligible fashion and written in standard English?

Reviewer #1: Yes

Reviewer #2: Yes

5. Review Comments to the Author

Reviewer #1: Manuscript Number: PONE-D-20-25658

Title: Targeted temperature management by inhibiting HMGB1 release in myocardial ischemia/reperfusion injury

Plos One

Reviewer’s comments:

The authors investigated the expression and release of HMGB1 as a possible mediator of the innate immune response after acute myocardial ischemia/reperfusion at normothermia, with targeted temperature management at 33°C and 36°C, and with Glycyrrhizin, a pharmacological HMGB1 inhibitor, at normothermia. They observed similar cytoprotective effect by TTM at both temperatures and also by treatment with Glycyrrhizin instead of cooling, as shown in the attenuation of apoptosis (TUNEL) and necrosis (infracted area and serum TnT), relative to normoxic control. Moreover, they also showed an attenuation of the MI-induced innate immune response (IL-6, TnF-a, and IL-1ß expressions) by the various treatments, presumably due to the attenuation of HMGB1 release into the extracellular matrix. However, the attenuation of HMGB1 release by the different treatments was not shown in the date presented. Although the paper shows valuable and significant data, some major and minor issues clearly need to be addressed.

Major issues:

1. The authors only showed the intracellular expression of HMGB1 by IF staining and no data on HMGB1 serum concentration. Therefore, the only conclusion that can be drawn is MI-induced injury resulted in decreased intracellular HMGB1 expression, and no conclusion can be made if HMGB1 has been released to the extracellular matrix. Similar to their assessment of serum TnT by ELISA, data on serum HMGB1 would greatly support their conclusions.

2. The authors do not address the effect of their cooling treatment on MI-induced apoptosis, although it is clearly supported by their presented data.

3. The authors assess the infarct volume optically in 2 mm thick slices, which may or may not contain the entire infarct volume and does not clearly represent the apoptotic and necrotic areas. No indication of a z-stack was mentioned for this infarct volume analysis.

4. The title is grammatically incorrect and does not correspond with the presented findings, which is confusing for the readers.

5. Significant differences in the data illustrated in the graphs needs to be shown more clearly to better let the readers see which groups are being compared.

6. Data for experimental groups with n=5 should be shown as mean ± standard deviation and not as mean ± standard error of the mean.

7. It is unclear which statistical analysis was used for each group comparison.

Minor Issues:

• Figure Legends - Figure 1B: It is not clear what the authors mean by, “Traces of rat body temperature”, when they mention measuring the core temperature.

• Methods – TUNEL: How many slices per group were used?

• It is unclear how the heart tissue for the assessment of infarct volume is between 0 and 8 cm from the apex and in the same time the material for the immunohistochemistry analysis also comes from between 4 and 6 mm from the apex of the heart. It would be also interesting to reveal why only the apex of the heart was taken instead of its proximal parts.

• It would also be of interest to see how the immune response would change in a sham cooled group.

• Why were the TTM groups excluded from Figure 5A, 5B and 5C?

Reviewer #2: In the present paper, Beom and colleagues showed that targeted temperature management (TTM) at 33℃ and 36℃ had similar protective effects in a myocardial ischemia/reperfusion (I/R) model. Specifically, they demonstrated that TTM significantly reduced infarct volume and attenuated the elevation of cardiac troponin, a sensitive and specific marker of heart muscle damage, after injury. Moreover, they described that both TTM at 33℃ and 36℃ equally inhibited the extracellular release of high mobility group box-1(HMGB1) from necrotic cardiomyocytes in infarct tissue and suppressed the expression of inflammatory cytokines from peri-infarct regions. The exact mechanism by which hypothermia attenuates myocardial damage due to ischemia and reperfusion remains unknown. For this reason, the Authors hypothesized that the cardioprotective effect of TTM was dependent on the inhibition of HMGB1 release, basing on the fact that I/R injury after a pre-treatment with glycyrrhizin (known as a pharmacological inhibitor of HMGB1 release) showed similar protective effects as TTM.

As the Authors declared, this is the first study to suggest that a target core temperature of 36°C is applicable for cardioprotection: this becomes important from a therapeutic point of view, considering that therapeutic hypothermia set at lower temperature commonly induces several harmful effects.

I think the present paper is suitable to be published in PlosOne even if in my opinion there are some specific issues that need to be addressed.

- The Authors described that “in the TTM groups with target temperatures of 33°C and 36℃, external surface cooling was started at 15 min after LAD coronary ligation by placing ice packs on the animal’s torso” and that “the core target temperatures of 33 ± 0.5°C and 36 ± 0.5℃ were reached within 18 and 8 min after the onset of TTM, respectively, and then maintained for 4 h”. So, as well represented in Figure 1B, in the 33°C group the target temperature was reached after the reperfusion had already started and this was not the case of the 36℃, where the target temperature was reached before reperfusion. Don’t the Authors think that this could invalidate the comparison between the two groups?

- The Authors reported that “when ischemic damage to the myocardium is induced by LAD ligation of the heart, HMGB1 is released from the nucleus of myocardial cells [30,31]”. In my opinion, to unequivocally demonstrate the HMGB1 release in the plasma, the Authors should perform an ELISA assay for the quantification of HMGB1 in plasma, because with immunofluorescence they demonstrated only a decrease that could be potentially also a consequence of a reduced expression after MI.

Moreover, several studies report an increase of myocardial HMGB1 levels in experimental models of I/R [https://doi.org/10.1161/CIRCU LATIONAHA.108.769331; https://doi.org/10.1093/cvr/cvq373], beside the increase of circulating HMGB1 levels derived from necrotic cardiomyocytes and active secretion by hypoxic cardiac and infiltrating inflammatory cells: in fact, myocardial HMGB1 expression increases soon after ischemia and remains high several days after reperfusion [https://doi.org/10.1152/ajpheart.00703.2010]. Did the Authors observe a similar increase in myocardial HMGB1 expression? And if not, did it may depend on the too much short time of reperfusion (perhaps 4hours is too early to see an increase of HMGB1 expression)? Or did it maybe depend onthe myocardial area? In my opinion, the results presented in this paper could realistically describe the myocardial area interested by the infarct (in which nuclear HMGB1 is almost completely released), but the Authors talked about “peri-infarcted area” that means, I suppose, the area on the border between the infarcted area and border zone, in which cardiomyocytes don’t die and, as a consequence, don’t release HMGB1, but on the contrary increase HMGB1 expression. I suggest the Authors to deeper discuss this point.

- Figure 2a: A lot of background signal is present both in the nuclei staining (blue) and in HMGB1 staining (red) in SHAM, MI+33 and, even if less, in MI+36, making the comparison among different groups very complicated. I suggest the Authors to replace the pictures representing the groups mentioned above, to provide a more convincing figure and to better reflect the graph in Figure 2b.

- Figures 2c,d,e and 5a,b,c: In the graphs representing the RT-PCR results, it’s better to set the value of SHAM groups at 1 and, as a consequence, correlate the graph bars of the other groups.

- The Authors demonstrated that in MI+33 and MI+36 there was a very weak release of HMGB1 (even less than in SHAM) and expression of inflammatory cytokines (even less than in SHAM for TNFα). Given that other studies demonstrated a great increase of plasma concentrations of HMGB1 with a peak 12h after MI and a marked decrease of HMGB1 levels in the infarct zone by Western Blot 24h after MI [https://doi:10.1093/cvr/cvn163], I suggest the Authors to justify the choice to perform a reperfusion time of only 4hours, because it is possible that HMGB1 release and inflammatory cytokine production are just delayed by the TTM and some differences between MI+33 and MI+36 can become evident after a more long reperfusion time.

- Figure 3a:A lot of background signal is present both in the nuclei staining (blue) and in TUNEL positive nuclei staining (green) especially in MI group, making the comparison among different groups very complicated. I suggest the Authors to replace the pictures representing the MI group, to provide a more convincing figure and to better reflect the graph in Figure 3b.Moreover, it is really debatable that the number of nuclei present in MI peri-infarct region is higher than in SHAM left ventricle.

- Figure 4a: I think the Authors forgot to put the pictures of the SHAM hearts.

- Figures 4c and 4d:I suggest the Authors to switch the figure 4c and 4d with the immunofluorescence pictures before the quantificationas in figure 2.

- Figure 4d:A lot of background signal is present both in the nuclei staining (blue) and in HMGB1 staining (red) in SHAM and in Gly groups, making the comparison among different groups very complicated. I suggest the Authors to replace the pictures representing the groups mentioned above, to provide a more convincing figure and to better reflect the graph in Figure 4c.

- Figure 5d: Given that the Authors divided the present study into two main experiments, I believe that is more linear to provide two separate graphs also for the measurement of serum cTnT (one in figure 2 and one in figure 5). This becomes important also considering that the values showed for the other parameters (e.g. inflammatory cytokine expression) resulted a bit different between the different SHAM and MI groups of the two sets of experiments.

- Lastly, the major point of my revision. The Authors divided the present study into two main experiments: the first one to assess the effects of myocardial protection through TTM at 33°C and 36℃ and the second one to verify the cardioprotective effect of TTM by specifically inhibiting HMGB1 release from cells with Glycyrrhizin. But in this way the Authors cannot provide the direct demonstration that the cardioprotective effect of TTM is mediated by the blocking of HMGB1 release. They can only speculate about it on the basis of the similar results obtained in the two main experiments. The correct demonstration of this should be with the use of inducers of HMGB1 release in I/R damage in conditions of TTM. I suggest the Authors to take the cue from [http://doi:10.1038/s41598-017-14848-1] for new pharmacological agents that induce HMGB1 release. Otherwise it’s just a speculation and they have to underline that in their Discussion session!

6. PLOS authors have the option to publish the peer review history of their article (what does this mean?). If published, this will include your full peer review and any attached files.

Reviewer #1: **Yes: **Giang Tong

Reviewer #2: No

---

## [Author Response · Author response to Decision Letter 0]

23 Nov 2020

please, check and review attached file - Reviewers' comments 

Response to reviewers’ comments

Reviewer #1: Major issues:

Response: Dear reviewer, we appreciate your helpful comments and suggestions regarding our manuscript.

1. The authors only showed the intracellular expression of HMGB1 by IF staining and no data on HMGB1 serum concentration. Therefore, the only conclusion that can be drawn is MI-induced injury resulted in decreased intracellular HMGB1 expression, and no conclusion can be made if HMGB1 has been released to the extracellular matrix. Similar to their assessment of serum TnT by ELISA, data on serum HMGB1 would greatly support their conclusions.

Response: Thank you for this helpful comment. As per your comment, to identify extracellular release of HMGB1, we conducted an additional experiment to measure the serum HMGB1 level by HMGB1 ELISA. We have added the results to the manuscript and restructured the manuscript based on the reviewer’s comment.

Enzyme-linked immunosorbent assay (ELISA) 

1. Enzyme-linked immunosorbent assay (ELISA) for cardiac troponin T (cTnT) and HMGB1

To obtain serum samples from rats, blood was drawn from the right atrium at 4 h after ligation of the LAD coronary artery with a 22-gauge needle. One millilitre of collected blood was transferred into a Z Serum Sep Clot Activator (Greiner Bioone, Kremsmunster, Austria), followed by centrifugation for 15 min at 3,000 rpm. The cTnT concentrations were determined using the cTnT ELISA kit (MBS2024997, MyBioSource, San Diego, CA, USA) and HMGB1 concentrations were determined using the Rat HMGB1 ELISA kit (Solarbio, Beijing, China).

Next, we performed an ELISA to measure HMGB1 levels in serum samples obtained at 4 h after the onset of ischemia. As expected, the level of circulating of HMGB1 was increased after I/R injury, but this increase was significantly attenuated by TTM at 33°C and 36°C.

(normothermic group after myocardial I/R, 367.08 ± 83.58, 33°C TTM after myocardial I/R, 67.15 ± 15.55 and 36°C TTM after myocardial I/R, 66.30 ± 7.43 pg/mL; P < 0.001)

(deletion) In addition, our previous study demonstrated that serum HMGB1 level was lower in the TTM group than the group without TTM, demonstrating the correlation between TTM and HMGB1

In addition, we demonstrated that the serum HMGB1 level was lower in the TTM group than in the group without TTM, revealing a correlation between TTM and HMGB1.

2. The authors do not address the effect of their cooling treatment on MI-induced apoptosis, although it is clearly supported by their presented data.

Response: We agree with this helpful comment. We have added this point to the Introduction and Discussion sections of the manuscript.

We found that TTM attenuated the increase in infarct size, apoptosis, extracellular release of HMGB1, and pro-inflammatory effect against the propagation of injury in rats with AMI.

The inflammatory response and apoptotic cell death are known to play important roles in the development of ischemic heart damage caused by myocardial I/R injury. Apoptosis is an important mechanism in I/R injury, and therapeutic hypothermia reduces apoptosis in myocytes. Therapeutic hypothermia-induced myocardial protection is significantly associated with beneficial modifications in apoptotic signal pathways.

We also demonstrated that TTM attenuated myocardial I/R-induced apoptosis. Temperatures of 33°C or 36℃ in TTM induced equivalent myocardial protection by attenuating apoptosis after I/R injury.

3. The authors assess the infarct volume optically in 2 mm thick slices, washich may or may not contain the entire infarct volume and does not clearly represent the apoptotic and necrotic areas. No indication of a z-stack was mentioned for this infarct volume analysis.

Response: Thank you for this helpful comment. To measure the total infarct volume, heart tissue between 0 and 8 mm from the apex of the heart was used to measure consistent areas in all tissues. Additionally, all heart tissues in the experiment included the site of LAD ligation and all injured areas below the ligation site. 

According to your comment, we re-measured the entire infarct volume. We measured the infarcted area in the anterior and posterior sides of each 2-mm-thick slice using ImageJ 1.48v software. To determine the infarct volume in each slice, the average value of the infarct area on the anterior and posterior sides was multiplied by the thickness (2 mm) [thickness × (top area + bottom area)/2]. In addition, the total infarct volume was calculated as the sum of the infarct volume per slice. 

4. The title is grammatically incorrect and does not correspond with the presented findings, which is confusing for the readers.

Response: According to your comment, we have revised the title.

Targeted temperature management at 33°C or 36℃ induces equivalent myocardial protection by inhibiting HMGB1 release in myocardial ischemia/reperfusion injury

5. Significant differences in the data illustrated in the graphs needs to be shown more clearly to better let the readers see which groups are being compared.

Response: Thank you for this helpful comment. We have revised the graphs based on your comment.

6. Data for experimental groups with n=5 should be shown as mean ± standard deviation and not as mean ± standard error of the mean.

Response: Thank you for this helpful comment. We have revised all data as the mean ± standard deviation based on your comment.

Statistical analysis

All experimental results are expressed as the mean ± standard error deviation of the mean. Statistical analyses were performed using unpaired t-test or by one-way analysis of variance (ANOVA) followed by Bonferroni post hoc tests for multiple comparisons between groups. Differences with P < 0.05 were considered as significant.

7. It is unclear which statistical analysis was used for each group comparison.

Response: We have discussed this point with our statisticians to confirm the statistical analysis. We have added this information to the manuscript and figure legends.

Statistical analysis

All experimental results are expressed as the mean ± standard deviation of the mean. Statistical analyses were performed using unpaired t-test or by one-way analysis of variance (ANOVA) followed by Bonferroni post hoc tests for multiple comparisons between groups. Differences with P < 0.05 were considered as significant.

Minor Issues:

• Figure Legends - Figure 1B: It is not clear what the authors mean by, “Traces of rat body temperature”, when they mention measuring the core temperature.

Response: Thank you for this helpful comment. We have revised figure legend as follows based on your comment.

Figure 1B. Changes in rat body temperature after LAD ligation

• Methods – TUNEL: How many slices per group were used?

Response: Thank you for this helpful comment. We have added this point to Methods as follows.

Methods

TUNEL assay

One slide from each animal was selected and stained. The two peri-ischemic areas of the stained sections were observed with a confocal microscope (LSM 700; Carl Zeiss GmbH, Jena, Germany). The average values of TUNEL-positive cells in the peri-infarct area were derived from two areas on the stained sections. 

• It is unclear how the heart tissue for the assessment of infarct volume is between 0 and 8 cm from the apex and in the same time the material for the immunohistochemistry analysis also comes from between 4 and 6 mm from the apex of the heart. It would be also interesting to reveal why only the apex of the heart was taken instead of its proximal parts.

Response: Thank you for this helpful comment. We have added this point to Methods as follows.

To measure the total infarct volume, heart tissue between 0 and 8 mm from the apex of the heart was used to measure consistent areas in all tissues. Additionally, all heart tissues in the experiment included the site of LAD ligation and all injured areas below the ligation site. The proximal part of the heart has a complicated structure with several blood vessels, the right atrium, and the left atrium. Therefore, it was difficult to identify an accurate reference point for cutting the tissue, whereas the apex had a simple and consistent structure, making it a better point as a standard for accurate tissue cutting. After TTC staining, we confirmed the infarct area and performed immunostaining by selecting a slide containing normal tissue (peri-ischemic area) around the infarct. TTC staining showed that the infarcted and normal tissues (peri-ischemic area) were distributed in the tissue between 4 and 6 mm from the apex. We used this area for immunohistochemistry analysis.

Methods

Immunohistochemistry analysis

For immunohistochemistry analysis, 2,3,5-TTC staining was performed to confirm the peri-infarct area in the left ventricle. Next, 2-mm-thick slices between 4 and 6 mm from the apex of the rat heart were selected, fixed with 4% paraformaldehyde solution, and embedded in paraffin. Between 4 and 6 mm from the apex of the rat heart was chosen because the peri-infarcted region was easily observable given that it was properly mixed with normal and infarct tissue after TTC staining.

• It would also be of interest to see how the immune response would change in a sham cooled group.

Response: Thank you for this helpful comment. According to your recommendations, we conducted the experiments to identify changes in the sham at target temperatures of 33°C and 36°C. We have added the results to the manuscript and restructured the manuscript based on the reviewer’s comment.

To assess the effects of myocardial protection exerted by TTM at 33°C and 36℃, the rats were randomly divided into four experimental groups: sham + 37°C (n = 5), sham + 33°C TTM (n = 5), sham + 36°C TTM (n = 5), LAD I/R + 37°C normothermia (n = 5), LAD I/R + 33°C TTM (n = 5), and LAD I/R + 36°C TTM (n = 5).

• Why were the TTM groups excluded from Figure 5A, 5B and 5C?

Response: To verify the cardioprotective effect of TTM by specifically inhibiting HMGB1 in our animal model, we divided the present study into two main experiments: (1) examination of the cardioprotective effect of TTM and (2) analysis of the cardioprotective effect of a pharmacologic inhibitor against HMGB1. 

We apologize for the confusion created by including the results of two experiments for the levels of cardiac troponin T (cTnT) in the plasma in Fig 5D to reduce the number of figures. We have separated the figure showing the measurement of serum cTnT into two figures.

Response to reviewers’ comments

Reviewer #2: 

Response: Dear reviewer, we appreciate your helpful comments and suggestions regarding our manuscript.

- The Authors described that “in the TTM groups with target temperatures of 33°C and 36℃, external surface cooling was started at 15 min after LAD coronary ligation by placing ice packs on the animal’s torso” and that “the core target temperatures of 33 ± 0.5°C and 36 ± 0.5℃ were reached within 18 and 8 min after the onset of TTM, respectively, and then maintained for 4 h”. So, as well represented in Figure 1B, in the 33°C group the target temperature was reached after the reperfusion had already started and this was not the case of the 36℃, where the target temperature was reached before reperfusion. Don’t the Authors think that this could invalidate the comparison between the two groups?

Response: Thank you for this helpful comment. We apologize for the confusion created by our error. As we set the target core temperature to 33°C ± 0.5°C and 36°C ± 0.5℃, the time to reach the target core temperature is described as follows: “the core target temperatures of 33 ± 0.5°C and 36 ± 0.5℃ were reached within 18 and 8 min after the onset of TTM, respectively, and then maintained for 4 h”. We have corrected this text as follows: “the core target temperatures of 33°C and 36℃ were reached within 18 ± 1.41 and 8 ± 0.80 min after the onset of TTM, respectively, and then maintained for 4 h. The core target temperatures of 33°C ± 0.5°C and 36°C ± 0.5℃ were reached within 13 ± 0.80 and 5 ± 0.49 min after the onset of TTM. The average values of the core temperature upon reperfusion were 33.2°C ± 0.07°C in the 33°C group and 35.8°C ± 0.05°C in the 36°C groups.”

- The Authors reported that “when ischemic damage to the myocardium is induced by LAD ligation of the heart, HMGB1 is released from the nucleus of myocardial cells [30,31]”. In my opinion, to unequivocally demonstrate the HMGB1 release in the plasma, the Authors should perform an ELISA assay for the quantification of HMGB1 in plasma, because with immunofluorescence they demonstrated only a decrease that could be potentially also a consequence of a reduced expression after MI.

Response: Thank you for this helpful comment. To detect extracellular release of HMGB1, we measured the serum HMGB1 level by HMGB1 ELISA. We have added the results to the manuscript and restructured the manuscript based on your comments.

(normothermic group after myocardial I/R, 367.08 ± 83.58, 33°C TTM after myocardial I/R, 67.15 ± 15.55 and 36°C TTM after myocardial I/R, 66.30 ± 7.43 pg/mL; P < 0.001)

Enzyme-linked immunosorbent assay (ELISA) 

2. Enzyme-linked immunosorbent assay (ELISA) for cardiac troponin T (cTnT) and HMGB1

2. To obtain serum samples from rats, blood was drawn from the right atrium at 4 h after ligation of the LAD coronary artery with a 22-gauge needle. One millilitre of collected blood was transferred into a Z Serum Sep Clot Activator (Greiner Bioone, Kremsmunster, Austria), followed by centrifugation for 15 min at 3,000 rpm. The cTnT concentrations were determined using the cTnT ELISA kit (MBS2024997, MyBioSource, San Diego, CA, USA) and HMGB1 concentrations were determined using the Rat HMGB1 ELISA kit (Solarbio, Beijing, China).

3. Next, we performed an ELISA to measure HMGB1 levels in serum samples obtained at 4 h after the onset of ischemia. As expected, the level of circulating of HMGB1 was increased after I/R injury, but this increase was significantly attenuated by TTM at 33°C and 36°C.

(normothermic group after myocardial I/R, 367.08 ± 83.58, 33°C TTM, 67.15 ± 15.55 and 36°C TTM, 66.30 ± 7.43 pg/mL, respectively; P < 0.001)

Moreover, several studies report an increase of myocardial HMGB1 levels in experimental models of I/R [https://doi.org/10.1161/CIRCU LATIONAHA.108.769331; https://doi.org/10.1093/cvr/cvq373], beside the increase of circulating HMGB1 levels derived from necrotic cardiomyocytes and active secretion by hypoxic cardiac and infiltrating inflammatory cells: in fact, myocardial HMGB1 expression increases soon after ischemia and remains high several days after reperfusion [https://doi.org/10.1152/ajpheart.00703.2010]. Did the Authors observe a similar increase in myocardial HMGB1 expression? And if not, did it may depend on the too much short time of reperfusion (perhaps 4hours is too early to see an increase of HMGB1 expression)? Or did it maybe depend on the myocardial area? In my opinion, the results presented in this paper could realistically describe the myocardial area interested by the infarct (in which nuclear HMGB1 is almost completely released), but the Authors talked about “peri-infarcted area” that means, I suppose, the area on the border between the infarcted area and border zone, in which cardiomyocytes don’t die and, as a consequence, don’t release HMGB1, but on the contrary increase HMGB1 expression. I suggest the Authors to deeper discuss this point.

Response: Thank you for this helpful comment. Firstly, we hypothesized that TTM could attenuate the inflammatory response in peri-infarct regions by inhibiting extracellular release of HMGB1 in a rat LAD coronary artery ligation model and subsequently reduce the myocardial infarcted area, resulting in increased myocardial protection after I/R injury. We investigated whether TTM at 36℃ has a myocardial protective effect via the same mechanism. 

HMGB1 is a ubiquitously and abundantly expressed non-histone DNA binding protein with dual functions. When in the nucleus, HMGB1 functions to stabilize the DNA structure and modulates transcriptional activity. However, when cells are damaged or activated, HMGB1 is released into the extracellular environment where it acts as an inflammatory cytokine that mediates cytokine release, inflammation, and endothelial activation. HMGB1 is an early mediator of sterile injury and late mediator of infection. During ischemia and other forms of sterile cell injury, HMGB1 is released as an early mediator that in turn activates the later release of TNF and other cytokines. Studies of animal models revealed that HMGB1 levels are significantly increased during ischemia-reperfusion injury, increase within 1 h after reperfusion, and remain elevated for up to 24 h. Nuclear HMGB1 translocates from the neuronal cell nucleus into the cytoplasm within 1 h after the onset middle cerebral artery occlusion and is then exported from the cell.

As our study focused on HMGB1 already present in cells, we provide evidence that TTM inhibits the propagation of ischemic damage by inhibiting the extracellular release of HMGB1 already present in cells. 

We conducted additional experiments according to the reviewers’ comments. 

Upon LAD coronary artery ligation-induced ischemic injury, HMGB1 is released from the heart cell nuclei, reducing the number of HMGB1-positive cells in the peri-ischemic area. We next determined the serum HMGB1 level. As expected, the level of circulating of HMGB1 was increased after I/R injury, but this increase was significantly attenuated by TTM. We performed RT-PCR to examine the mRNA levels of HMGB1 in the peri-infarct region at 4 h after I/R injury and found no significant increase in the mRNA levels of HMGB1 of all groups by 4 h after I/R injury. Based on the mRNA expression of HMGB1, HMGB1 expression did not appear to be increased at 4 h after injury. It may take time for the new HMGB1 protein to be produced in the cell after the extracellular release of HMGB1 already present in cells. 

To determine the specific inhibition effects of HMGB1 on ischemic injury, in a previous study, we injected a neutralizing antibody against HMGB1 into the intracerebroventricular space of rats. TTC staining after 4 h of ischemia showed that HMGB1 neutralizing antibody treatment reduced the MCAO-induced cortical infarct volume. This result supports our hypothesis that HMGB1 inhibition effectively protects the brain against the spread of ischemic injury. We also found that glycyrrhizin as a pharmacological HMGB1 inhibitor showed similar effects by inhibiting the extracellular release of HMGB1 compared with HMGB1 neutralizing antibody. Considering the characteristics of the heart where neutralizing antibodies cannot be injected locally and systemic injection of neutralizing antibodies causes problems, we could not evaluate the protective effects using neutralizing antibodies against HMGB1. Instead, we used glycyrrhizin rather than an HMGB1 neutralizing antibody. Both TTM and glycyrrhizin attenuate the spread of ischemic injury in peri-infarct regions by inhibiting the extracellular release of HMGB1 in a rat LAD coronary artery ligation model. 

According to the reviewer’s comment, myocardial HMGB1 expression increases soon after ischemia and remains high for several days after reperfusion. However, we found s no significant increase in the mRNA levels of HMGB1 of all groups in 4 h after I/R injury.

(normothermic group after myocardial I/R, 1.33 ± 0.52, 33°C TTM after myocardial I/R, 1.06 ± 0.30, 36°C TTM after myocardial I/R, 0.63 ± 0.18, respectively; P = 0.20 (not significant))

 HMGB1 plays dual roles in cardiac injury. In the initial stage of cerebrovascular and cardiovascular diseases, HMGB1 is released into the outside of the cell to participate in the cascade amplification reaction of inflammation, causing vasospasm and apoptosis. In the recovery stage of disease, HMGB1 can promote tissue repair, regeneration, and remodelling. 

It would be appreciated if you could consider the role of extracellular release of HMGB1 that already exists in cells and role of HMGB1 that is newly generated after ischemic injury. Based on the role of HMGB1 already present in cells and released into the extracellular environment, 4 h appears to be an appropriate time for confirming the beneficial effect of TTM after injury.

We agree that identifying the role of newly generated HMGB1 after ischemic injury is an important topic of further study. Studies are needed to investigate whether newly generated HMGB1 is involved in inducing inflammatory reactions or repairs damage over time after injury.

Based on your comment, we have added this point to the Discussion section of the manuscript. 

HMGB1 levels are significantly increased within 1 h and maintained for up to 24 h during I/R injury. Therefore, it is important to prevent the action of HMGB1 to alleviate ischemic injury of the myocardium. Nevertheless, HMGB1 plays dual roles in cardiac injury. In the initial stage of cerebrovascular and cardiovascular diseases, HMGB1 is released from the cell to participate in the cascade amplification reaction of inflammation, causing vasospasm and apoptosis. In the recovery stage of disease, HMGB1 can promote tissue repair, regeneration, and remodelling. It is necessary to investigate whether the newly generated HMGB1 plays a role in propagating inflammatory detriment or repairing damages over time after injury.

Additional References

1. Kim JB, Lim CM, Yu YM, Lee JK. Induction and subcellular localization of high-mobility group box-1 (HMGB1) in the postischemic rat brain. J Neurosci Res. 2008;86: 1125-1131. doi:10.1002/jnr.21555 PMID:17975839

2. Andersson U, Tracey KJ. HMGB1 is a therapeutic target for sterile inflammation and infection. Annu Rev Immunol. 2011;29: 139-162. doi:10.1146/annurev-immunol-030409-101323 PMID:21219181

3. Lee JH, Yoon EJ, Seo J, Kavoussi A, Chung YE, Chung SP, et al. Hypothermia inhibits the propagation of acute ischemic injury by inhibiting HMGB1. Mol Brain. 2016;9: 81. doi:10.1186/s13041-016-0260-0 PMID:27544687

4. Qiu J, Nishimura M, Wang Y, Sims JR, Qiu S, Savitz SI, et al. Early release of HMGB-1 from neurons after the onset of brain ischemia. J Cereb Blood Flow Metab. 2008;28: 927-938. doi:10.1038/sj.jcbfm.9600582 PMID:18000511

- Figure 2a: A lot of background signal is present both in the nuclei staining (blue) and in HMGB1 staining (red) in SHAM, MI+33 and, even if less, in MI+36, making the comparison among different groups very complicated. I suggest the Authors to replace the pictures representing the groups mentioned above, to provide a more convincing figure and to better reflect the graph in Figure 2b.

Response: Thank you for this helpful comment. Based on your recommendations, we have replaced the image representing the sham and MI+33°C groups with more convincing images and revised the graph in Fig 2b.

- Figures 2c,d,e and 5a,b,c: In the graphs representing the RT-PCR results, it’s better to set the value of SHAM groups at 1 and, as a consequence, correlate the graph bars of the other groups.

Response: Thank you for this helpful comment. We have revised the graphs in Figure 2 and 5. 

- The Authors demonstrated that in MI+33 and MI+36 there was a very weak release of HMGB1 (even less than in SHAM) and expression of inflammatory cytokines (even less than in SHAM for TNFα). Given that other studies demonstrated a great increase of plasma concentrations of HMGB1 with a peak 12h after MI and a marked decrease of HMGB1 levels in the infarct zone by Western Blot 24h after MI [https://doi:10.1093/cvr/cvn163], I suggest the Authors to justify the choice to perform a reperfusion time of only 4hours, because it is possible that HMGB1 release and inflammatory cytokine production are just delayed by the TTM and some differences between MI+33 and MI+36 can become evident after a more long reperfusion time.

Response: Thank you for this helpful comment. In a study using HMGB1 Tg mice, Kitahara et al. demonstrated the first in vivo evidence that HMGB1 enhances angiogenesis, restores cardiac function, and improves survival after MI. Plasma HMGB1 levels after MI were significantly increased in both Wt and HMGB1-Tg mice, showing a peak at 12 h after MI. Particularly, the plasma HMGB1 level in HMGB1-Tg mice at 24 h after MI was significantly increased compared to that in Wt mice. These data suggest that large amounts of HMGB1 were released into the circulation from necrotic cardiomyocytes in HMGB1-Tg mice compared to in Wt mice. Our study demonstrated that TTM inhibits the propagation of ischemic damage by inhibiting the extracellular release of HMGB1 already present in cells. 

HMGB1 plays dual roles in cardiac injury. In the initial stage of cardiovascular diseases, HMGB1 is released from the cell to participate in the cascade amplification reaction of inflammation, causing vasospasm and apoptosis. In the recovery stage of disease, HMGB1 can promote tissue repair, regeneration, and remodelling. It would be appreciated if you could consider the role of extracellular release of HMGB1 that already exists in nuclei of cells and the role of HMGB1 that is newly generated after ischemic injury. Based on the study by Kitahara et al., newly generated HMGB1 with a peak at 12 h after ischemic injury exerts effects on recovery. Our study focused on inflammatory aggravation in the initial stage after ischemic injury. Considering the role of extracellular release of HMGB1 that already exists in cells, 4 h appears to be an appropriate time for confirming the beneficial effect of TTM by extracellular release of HMGB1 that already exists in cells after injury. However, we agree that identifying the role of newly generated HMGB1 after ischemic injury is an important topic requiring further study. To identify the role of HMGB1 after a longer time following I/R, it is necessary to investigate whether newly generated HMGB1 plays a role in inducing the inflammatory reaction or repairing damages over time after injury.

- Figure 3a: A lot of background signal is present both in the nuclei staining (blue) and in TUNEL positive nuclei staining (green) especially in MI group, making the comparison among different groups very complicated. I suggest the Authors to replace the pictures representing the MI group, to provide a more convincing figure and to better reflect the graph in Figure 3b. Moreover, it is really debatable that the number of nuclei present in MI peri-infarct region is higher than in SHAM left ventricle.

Response: Thank you for this helpful comment. Based on your comment, we have replaced the image showing the MI group to a more convincing picture and revised the graph in Fig 3b. In addition, we have replaced the image of the sham group to the other image of the sham group according to your comment. 

- Figure 4a: I think the Authors forgot to put the pictures of the SHAM hearts.

Response: Thank you for this helpful comment. We have added an image of the heart from the sham group to Fig 4a. 

- Figures 4c and 4d: I suggest the Authors to switch the figure 4c and 4d with the immunofluorescence pictures before the quantifications in figure 2.

Response: Thank you for this helpful comment. We have changed the layout of Fig 4.

- Figure 4d: A lot of background signal is present both in the nuclei staining (blue) and in HMGB1 staining (red) in SHAM and in Gly groups, making the comparison among different groups very complicated. I suggest the Authors to replace the pictures representing the groups mentioned above, to provide a more convincing figure and to better reflect the graph in Figure 4c.

Response: Thank you for this helpful comment. We have replaced the image representing the sham and Gly groups to more convincing pictures and revised the graph in Fig 4d.

- Figure 5d: Given that the Authors divided the present study into two main experiments, I believe that is more linear to provide two separate graphs also for the measurement of serum cTnT (one in figure 2 and one in figure 5). This becomes important also considering that the values showed for the other parameters (e.g. inflammatory cytokine expression) resulted a bit different between the different SHAM and MI groups of the two sets of experiments.

Response: Thank you for this helpful comment. We have separated the figure of the measurement of serum cTnT into two figures. 

- Lastly, the major point of my revision. The Authors divided the present study into two main experiments: the first one to assess the effects of myocardial protection through TTM at 33°C and 36℃ and the second one to verify the cardioprotective effect of TTM by specifically inhibiting HMGB1 release from cells with Glycyrrhizin. But in this way the Authors cannot provide the direct demonstration that the cardioprotective effect of TTM is mediated by the blocking of HMGB1 release. They can only speculate about it on the basis of the similar results obtained in the two main experiments. The correct demonstration of this should be with the use of inducers of HMGB1 release in I/R damage in conditions of TTM. I suggest the Authors to take the cue from [http://doi:10.1038/s41598-017-14848-1] for new pharmacological agents that induce HMGB1 release. Otherwise it’s just a speculation and they have to underline that in their Discussion session.

Response: Thank you for this helpful comment. This comment provides insight into the experimental methods used in our study. However, the study by Liu et al. is still in the screening stage of pharmacological agents that induce HMGB1 release, and it has been difficult to conduct experiments using the recommended agents because there are many factors that must be considered, such as the choice of various agents, effective dose, and interaction between TTM and agents. According to your comment, we have added this point to the Discussion as follows.

Liu et al. identified several classes of agents that potently induce the nucleo-cytoplasmic relocation and subsequent cellular release of HMGB1. To clarify whether TTM can attenuate the inflammatory response in peri-infarct regions by inhibiting the extracellular release of HMGB1 after I/R injury, further studies are needed to verify ischemic damage using pharmacological agents that induce HMGB1 release during TTM after I/R injury in the AMI model.

---

## [Decision Letter · Decision Letter 1]

23 Dec 2020

PONE-D-20-25658R1

Targeted temperature management at 33°C or 36℃ induces equivalent myocardial protection by inhibiting HMGB1 release in myocardial ischemia/reperfusion injury

PLOS ONE

Dear Dr. Je Sung You,

Thank you for submitting your manuscript to PLOS ONE. After careful consideration, we feel that it has merit but does not fully meet PLOS ONE’s publication criteria as it currently stands. Therefore, we invite you to submit a revised version of the manuscript that addresses the points raised during the review process.

In particular, the authors should revise the discussion as mentioned by one of the reviewer.

We look forward to receiving your revised manuscript.

Kind regards,

Federica Limana

Academic Editor

PLOS ONE

Journal Requirements:

Additional Editor Comments (if provided):

Reviewers' comments:

Reviewer's Responses to Questions

**Comments to the Author**

1. If the authors have adequately addressed your comments raised in a previous round of review and you feel that this manuscript is now acceptable for publication, you may indicate that here to bypass the “Comments to the Author” section, enter your conflict of interest statement in the “Confidential to Editor” section, and submit your "Accept" recommendation.

Reviewer #2: (No Response)

2. Is the manuscript technically sound, and do the data support the conclusions?

Reviewer #2: Partly

3. Has the statistical analysis been performed appropriately and rigorously? 

Reviewer #2: Yes

4. Have the authors made all data underlying the findings in their manuscript fully available?

Reviewer #2: Yes

5. Is the manuscript presented in an intelligible fashion and written in standard English?

Reviewer #2: Yes

6. Review Comments to the Author

Reviewer #2: Comments to Authors after paper revision

The Authors did their best to replay to all my comments on their article and I’m quite satisfied of their revision: new figures showed better quality, the results conserning missing experimental groups were provided, they performed ELISA assay for HMGB1 in serum, and I appreciated their care in clarifying my doubts about time of reperfusion and timing of HMGB1 release. I think that the present article had considerably improve its quality after the revision. For this reason, I agree for the publication, even if, in my opinion is still compulsory to better clarify the question of Glycyrrhizin. I understood the impossibility for the Authors to conduct experiments using pharmacological agents that induce HMGB1 release, but I think that is still not sufficiently clear in the paper that the second part of experiments (the part with Glycyrrhizin) did not provide the direct demonstration that the cardioprotective effect of TTM is mediated by the blocking of HMGB1 release. In fact, in my opinion, the blocking of HMGB1 release could be just a consequence and not be directly dependent on TTM. When the Authors compared Glycyrrhizin treatment with TTM, is just a speculation: the two treatments have the same final effect (myocardial protection during I/R), and present the same phenomenon (inhibition of HMGB1 release), but the mechanisms might be completely different. Glycyrrhizin impedes for sure HMGB1 release from cardiomyocytes because is a pharmacological inhibitor, TTM, on the contrary, might potentially act on some other factors that merge on the blocking of HMGB1 release. The effects are similar, but the mechanisms just could be. And the experiments with Glycyrrhizin did not provide any evidence on it; they only showed similarity between the two treatments and the Authors are just speculating, not demonstrating anything on this point. This has to be clear in order for the publication of the present paper.

7. PLOS authors have the option to publish the peer review history of their article (what does this mean?). If published, this will include your full peer review and any attached files.

Reviewer #2: No

---

## [Author Response · Author response to Decision Letter 1]

2 Jan 2021

**please, check the attached file for Response to Reviewer’s Comment **

Response to Reviewer’s Comment 

Response: Thank you for your helpful comment; we agree with it. Per your comment, we also think that the inhibition of HMGB1 release may be one of the mechanisms underlying the cardioprotective action of TTM and may result from several protective effects of TTM after I/R injury. Although the experiments involving glycyrrhizin and neutralizing antibodies only demonstrated that the extracellular release of HMGB1 is a key factor in the cardiac damage after I/R injury, there is no evidence that the blockade of the HMGB1 release is direct mechanism of the TTM-induced cardiac protection after I/R injury. We have mentioned this point in the Discussion section of the revised manuscript.

Discussion

We demonstrated that the serum HMGB1 level was lower in the TTM group than in the group without TTM, revealing a correlation between TTM and HMGB1. However, our study lacks data regarding the link between HMGB1 and I/R injury and on the direct mechanisms underlying this association. In the case of the AMI animal model, we could not confirm the relationship between HMGB1 and I/R injury because HMGB1-neutralizing antibodies could not be injected locally. Instead, we indirectly revealed the association of HMGB1 with I/R injury by using glycyrrhizin as a pharmacological HMGB1 inhibitor and via experiments using our middle cerebral artery occlusion model [24, 26]. However, these analyses with glycyrrhizin and neutralizing antibodies only demonstrated that the extracellular release of HMGB1 is a key factor in the cardiac damage after I/R injury. There is no evidence that the blockade of the HMGB1 release is direct mechanism of the TTM-induced cardiac protection after I/R injury. Further studies are needed to identify the direct mechanisms underlying, and the link between, HMGB1 release and the action of TTM after I/R injury. In addition, Liu et al. have identified several classes of agents that potently induce the nucleo-cytoplasmic relocation and subsequent cellular release of HMGB1 [45]. To clarify whether TTM can attenuate the inflammatory response in peri-infarct regions by inhibiting the extracellular release of HMGB1 after I/R injury, further studies are needed to verify the ischemic damage by using pharmacological agents that induce HMGB1 release during TTM after I/R injury in the AMI model.

---

## [Decision Letter · Decision Letter 2]

13 Jan 2021

Targeted temperature management at 33°C or 36℃ induces equivalent myocardial protection by inhibiting HMGB1 release in myocardial ischemia/reperfusion injury

PONE-D-20-25658R2

Dear Dr. Je Sung You,

We’re pleased to inform you that your manuscript has been judged scientifically suitable for publication and will be formally accepted for publication once it meets all outstanding technical requirements.

Kind regards,

Federica Limana

Academic Editor

PLOS ONE

Additional Editor Comments (optional):

Reviewers' comments:

Reviewer's Responses to Questions

**Comments to the Author**

1. If the authors have adequately addressed your comments raised in a previous round of review and you feel that this manuscript is now acceptable for publication, you may indicate that here to bypass the “Comments to the Author” section, enter your conflict of interest statement in the “Confidential to Editor” section, and submit your "Accept" recommendation.

Reviewer #2: All comments have been addressed

2. Is the manuscript technically sound, and do the data support the conclusions?

Reviewer #2: Yes

3. Has the statistical analysis been performed appropriately and rigorously? 

Reviewer #2: Yes

4. Have the authors made all data underlying the findings in their manuscript fully available?

Reviewer #2: Yes

5. Is the manuscript presented in an intelligible fashion and written in standard English?

Reviewer #2: Yes

6. Review Comments to the Author

Reviewer #2: In my opinion, with the update that Authors introduced in the Discussion Section, the present paper is now suitable for the publication in PLOS ONE.

7. PLOS authors have the option to publish the peer review history of their article (what does this mean?). If published, this will include your full peer review and any attached files.

Reviewer #2: No

---

## [Editor Report · Acceptance letter]

18 Jan 2021

PONE-D-20-25658R2 

Targeted temperature management at 33°C or 36℃ induces equivalent myocardial protection by inhibiting HMGB1 release in myocardial ischemia/reperfusion injury 

Dear Dr. You:

I'm pleased to inform you that your manuscript has been deemed suitable for publication in PLOS ONE. Congratulations! Your manuscript is now with our production department. 

Kind regards, 

on behalf of

Dr. Federica Limana 

Academic Editor

PLOS ONE